# Microglial neuropilin-1 promotes oligodendrocyte expansion during development and remyelination by trans-activating platelet-derived growth factor receptor

Amin Sherafat[1], Friederike Pfeiffer[1,2], Alexander M. Reiss[1], William M. Wood[1] & Akiko Nishiyama [1,3,4 ✉]

Nerve-glia (NG2) glia or oligodendrocyte precursor cells (OPCs) are distributed throughout the gray and white matter and generate myelinating cells. OPCs in white matter proliferate more than those in gray matter in response to platelet-derived growth factor AA (PDGF AA), despite similar levels of its alpha receptor (PDGFRα) on their surface. Here we show that the type 1 integral membrane protein neuropilin-1 (Nrp1) is expressed not on OPCs but on amoeboid and activated microglia in white but not gray matter in an age- and activity-dependent manner. Microglia-specific deletion of Nrp1 compromised developmental OPC proliferation in white matter as well as OPC expansion and subsequent myelin repair after acute demyelination. Exogenous Nrp1 increased PDGF AA-induced OPC proliferation and PDGFRα phosphorylation on dissociated OPCs, most prominently in the presence of sub-optimum concentrations of PDGF AA. These findings uncover a mechanism of regulating oligodendrocyte lineage cell density that involves trans-activation of PDGFRα on OPCs via Nrp1 expressed by adjacent microglia.

[1] Department of Physiology and Neurobiology, University of Connecticut, Storrs, CT, USA. [2] Department of Physiology, University of Tübingen, Tübingen, Germany. [3] Institute for Systems Genomics, University of Connecticut, Storrs, CT, USA. [4] The Connecticut Institute for the Brain and Cognitive Sciences, University of Connecticut, Storrs, CT, USA. ✉email: akiko.nishiyama@uconn.edu

Oligodendrocyte precursor cells (OPCs), also known as NG2 glia or polydendrocytes, are widely distributed throughout the developing and mature central nervous system (CNS)[1–3], where they comprise 2–9% of the total cells. The developmental expansion of this population is critically dependent on platelet-derived growth factor AA (PDGF AA) acting on their alpha receptor (PDGFRα)[4–6]. The most characterized and established role of OPCs is their ability to self-renew and generate the correct number of myelinating oligodendrocytes (OLs) needed for optimal network function. Hypomyelinating or dysmyelinating mouse mutants with reduced OLs or myelin are accompanied by increased OPC proliferation[7,8]. In the mature CNS, a demyelinated lesion rapidly elicits OPC proliferation[9–12]. However, the molecular and cellular mechanism by which such feedback signals promote OPC proliferation is not known.

While OPCs are evenly distributed throughout the CNS, the rate of OPC proliferation and OL differentiation is greater in white matter than in gray matter[13,14], and those in white matter proliferate more in response to PDGF AA[15]. While both cell intrinsic and extrinsic factors influence the differential OPC behavior[16], heterotopic transplantation of 300-μm³ pieces in slice cultures suggests that the greater proliferative response of OPCs in white matter to PDGF AA is imparted by their local pericellular microenvironment[15].

To determine the mechanism underlying the differential response of OPCs in gray and white matter to PDGF AA, we searched the literature for a potential co-receptor for PDGF that could differentially regulate the proliferative response of OPCs to PDGF AA in gray and white matter. We identified heparan sulfate proteoglycans (HSPGs), and neuropilin-1 (Nrp1)[17,18] as potential candidates. Since heparin and HSPGs are known to affect a variety of growth factors, we chose to explore the role of Nrp1 as a molecule with more specific targets. Nrp1 is a type 1 transmembrane molecule, which was first discovered in the developing *Xenopus* optic tectum[19] and subsequently shown to be expressed on developing murine axons[20] and regulate axon pathfinding by binding to class III semaphorins[21]. In endothelial cells, Nrp1 binds vascular endothelial cell growth factor (VEGF) and modulates signal transduction through VEGF receptor (VEGFR)[22]. Here, we show that Nrp1 is expressed on ameboid and activated microglia in the developing and demyelinated corpus callosum and promotes OPC proliferation by activating PDGFRα on OPCs in trans.

## Results

### Nrp1 modulates PDGF AA-dependent OPC proliferation in white matter.
To determine whether the proliferative response of OPCs to PDGF AA was modulated by Nrp1, we tested the effects of anti-Nrp1 antibody on PDGF AA-induced OPC proliferation in forebrain slice cultures from postnatal day 8 (P8) NG2cre;Z/EG double transgenic mice[15] (Fig. 1A). In the presence of 50 ng/mL PDGF AA alone, 42% of EGFP + OPCs in the corpus callosum were EdU + (Fig. 1D, F). By contrast only 18.4% of those in the cortex were EdU+ in the presence of PDGF AA (Fig. 1B, F). Anti-Nrp1 antibodies elicited a dose-dependent decrease in the proportion of EGFP + cells that proliferated in response to PDGF AA in the corpus callosum (Fig. 1F), but there was no effect on basal OPC proliferation in gray matter (Fig. 1C, F). We did not observe any increase in TUNEL + cells in the anti-Nrp1 antibody-treated slices (Fig. 1G, H) compared to slices treated with a control IgG, indicating that the antibody did not cause significant cell death. Thus, Nrp1 appeared to be necessary for PDGF AA-mediated OPC proliferation in the corpus callosum.

### Nrp1 is expressed on ameboid microglia in the developing corpus callosum.
To determine the relevant source of Nrp1 that

could modulate PDGF AA-dependent OPC proliferation in the corpus callosum, we examined the developing mouse CNS tissues for Nrp1 expression. In P5 brain, Nrp1 was detected widely in the forebrain (Fig. 2A), including vascular cells (arrowheads in Fig. 2A, C) along laminin+ blood vessels (Fig. S1A), as expected from previous reports on Nrp1 on vascular endothelial cells[22]. Nrp1 immunoreactivity was also detected on axons in the dorsal spinal cord (Fig. S1B) in P5 mice and in the dorsal corpus callosum in E18.5 brain (Fig. S1C), as previously shown (Kawakami et al. 1996). In addition, there was strong cellular staining in a subregion of the corpus callosum (Fig. 2A, boxed area magnified in Fig. 2D–F). The majority of the round Nrp1+ cells in the corpus callosum also expressed the microglial cell surface antigen F4/80[23] and had the appearance of ameboid microglia (Fig. 2A–F). Similarly, we detected Nrp1 on a subpopulation of F4/80+ ameboid microglia in P5 cerebellar white matter (not shown). We did not detect Nrp1 on PDGFRα+ OPCs (Fig. 2G–I), although Nrp1+ microglial processes were closely apposed to OPC cell bodies and processes (arrows in Fig. 2I). The close proximity to Nrp1-expressing ameboid microglia to OPCs suggested that microglial Nrp1 in the white matter could be modulating the proliferative response of OPCs to PDGF AA.

To further examine the enrichment of Nrp1 among microglia in white matter, we examined the expression of Nrp1 on EYFP + cells in the neocortex and corpus callosum of Cx3Cr1[creERT2-ires-EYFP] mice from P5 to P30 (Fig. 2J–M and Fig. S2). At P5 and P8, Nrp1 was expressed on the majority of EYFP+ ameboid microglia in the corpus callosum (97.8% at P5 and 98.4% at P8), but the enrichment declined by P14 to 14.0% (Fig. 2M), and Nrp1 was no longer detected at P30, as the EYFP+ microglia in the corpus callosum displayed progressively more ramified morphology. In the cortex, EYFP+ microglia had ramified morphology from P5 through P30 and did not co-express Nrp1. Thus, Nrp1 was preferentially expressed on ameboid cells in the early postnatal white matter.

The ameboid microglia that were found in P5 corpus callosum also expressed CD68 (Fig. S3, A–D, arrows), which is a lysosomal marker[24,25] and hence an indicator of phagocytic activity. The microglia in P14 corpus callosum exhibited a very small amount of punctate immunoreactivity for CD68, which was significantly lower compared to that in P5 corpus callosum, and these microglia were Nrp1-negative (Fig. S3, E–H, arrowheads).

### Microglia-specific Nrp1 deletion reduces developmental OPC proliferation in white matter.
To examine the effects of microglial Nrp1 on OPC proliferation, we deleted Nrp1 specifically from microglia using Nrp1 conditional knockout mice[26] crossed to Cx3CR1[creERT2-ires-EYFP] mice, which express tamoxifen-inducible cre specifically in Cx3CR1-expressing microglia[27] (Fig. 3A). When we induced Nrp1 deletion in microglia in mg-Nrp1-cont (fl/+) and mg-Nrp1-cko (fl/fl) mice by injecting 4-hydroxytamoxifen (4-OHT) at P2 and P3 and analyzed at P5, none of the EYFP+ microglia in mg-Nrp1 cko mice had detectable Nrp1 expression in the corpus callosum, whereas Nrp1 was detected on >99% of EYFP+ microglia in mg-Nrp1 cont heterozygous mice (Fig. S4A–D). Microglial Nrp1 deletion did not affect the density of microglia or astrocytes (Fig. S4E, F). Nrp1 immunoreactivity on blood vessels remained detectable in mg-Nrp1-cko (Fig. S4C, arrow), and the relative area in the corpus callosum occupied by blood vessels was not altered in mg-Nrp1-cko (Fig. S4G).

Having confirmed microglia-specific deletion of Nrp1 with this model, we next examined OPC proliferation in the corpus callosum of mg-Nrp1-cont and cko mice by EdU pulse labeling at P5 after microglial Nrp1 deletion at P2 and P3 (Fig. 3B). Comparison of the density of PDGFRα+ EdU+ cells in mg-

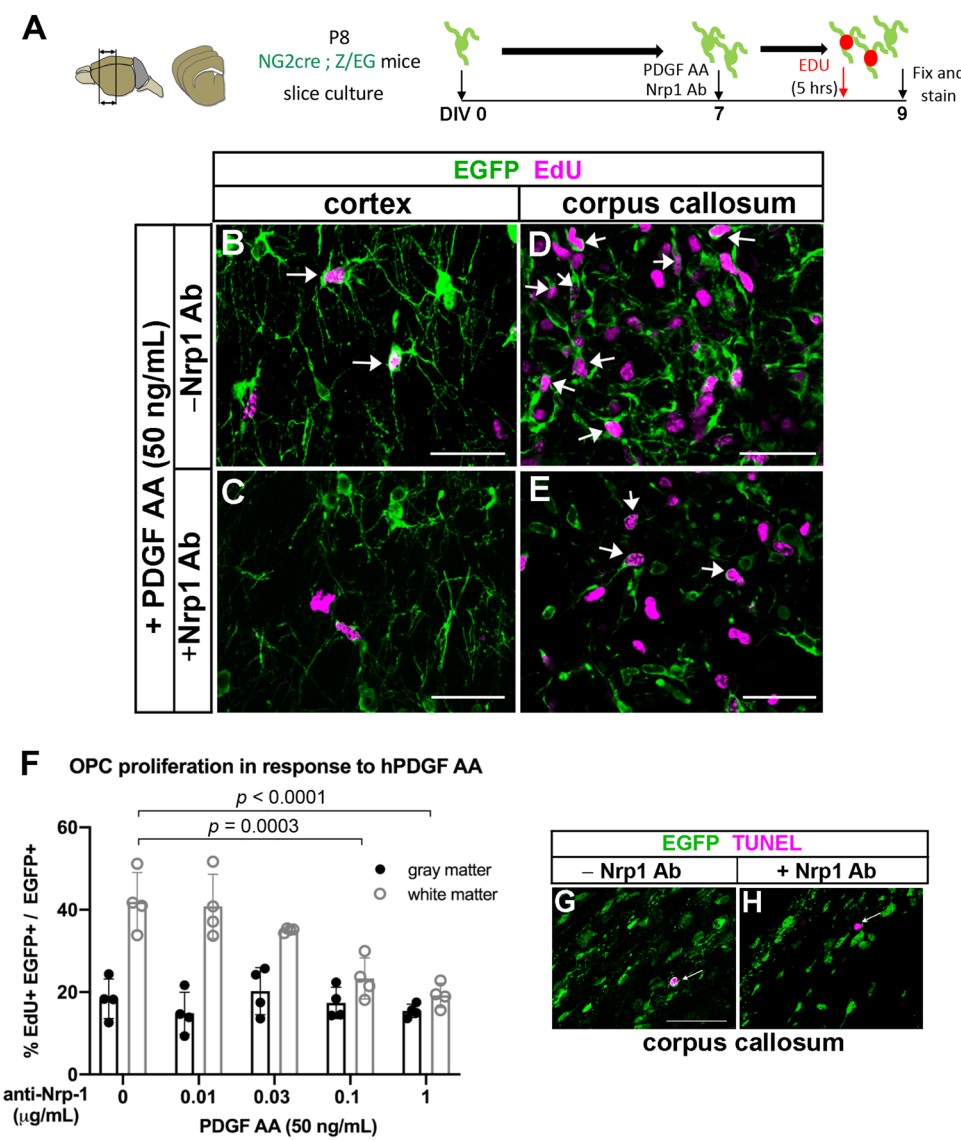

**Fig. 1 Anti-Nrp1 antibody blocks PDGF AA-mediated OPC proliferation in white but not gray matter. A** Schematic showing slice cultures from P8 NG2cre;Z/EG mice to assay for OPC proliferation in response to 50 ng/mL PDGF AA and different concentrations of anti-Nrp1 antibody. **B–E** Slice cultures from P8 NG2cre;Z/EG mice that were fixed and labeled for EGFP (green) and EdU (magenta) in the cortex (**B** and **C**) or corpus callosum (**D** and **E**) in the presence of 50 ng/mL PDGF AA in the absence (**B** and **D**) or presence (**C** and **E**) of 1 μg/mL anti-Nrp1 antibody. Arrows, examples of EGFP+ EdU+ cells. Scale bars, 50 μm. **F** Quantification of OPC proliferation in slice cultures after 5 h of EdU labeling. y-axis, proportion of EGFP + cells that were EdU+. Tukey's multiple comparisons test, $n = 4$, $F(4, 30) = 9.966$, means ± standard deviations. The differences between OPC proliferation in gray and white matter were significant at anti-Nrp1 antibody concentrations of 0 ($p < 0.0001$) and 0.01 μg/mL ($p = 0.0065$), but not at higher concentrations of the antibody. Black, gray matter; gray, white matter. Two-way ANOVA, Tukey's multiple comparisons test, $F(1, 30) = 89.71$, $n = 4$. **G, H** Slice cultures treated with 1 μg/mL control goat IgG (**G**) or goat anti-Nrp1 antibody (**H**) and labeled for EGFP (green) and TUNEL (magenta). Arrows indicate a TUNEL+ cell. Scale bars, 50 μm. Source data are provided as a Source Data file.

Nrp1-cont and cko corpus callosum revealed a 2.6-fold reduction in cko (Fig. 3C, D, E1). Since we observed that OPCs were closely apposed to Nrp1+ microglia in P5 corpus callosum as described above (Fig. 2G–I), we investigated whether OPCs that were in contact with Nrp1+ microglia were proliferating more than OPCs without microglial contact. We used 3D rotation of series of ≧40 confocal z-slices from P5 sections labeled for PDGFRα, Nrp1, EYFP, and EdU and assessed whether each PDGFRα+ OPC had an EYFP+ microglial element directly apposed to it and whether it had incorporated EdU (Fig. 3C, D, arrows; Fig. S5). Among the proliferating OPCs, 95.9% in mg-Nrp1-cont and 92.8% in mg-Nrp1-cko had contact with microglia.

In mg-Nrp1-cont corpus callosum, the contacts often involved intertwining processes of a PDGFRα+ OPC with microglia that expressed Nrp1 primarily on their processes. When we saw a direct apposition of microglia and OPC by confocal microscopy, we scored the OPC as having contact (Fig. S5A–G). When we did not detect direct apposition through all the z-stacks we scored them as having no contact (Fig. S5H–N). There was no significant difference in the density of proliferating OPCs that were not in contact with microglia between the two genotypes (Fig. 3E2, triangles). By contrast, the density of proliferating OPCs that were in contact with microglia was 2.7-fold lower in mg-Nrp1-cko compared to cont (Fig. 3E, circles). This indicated that the

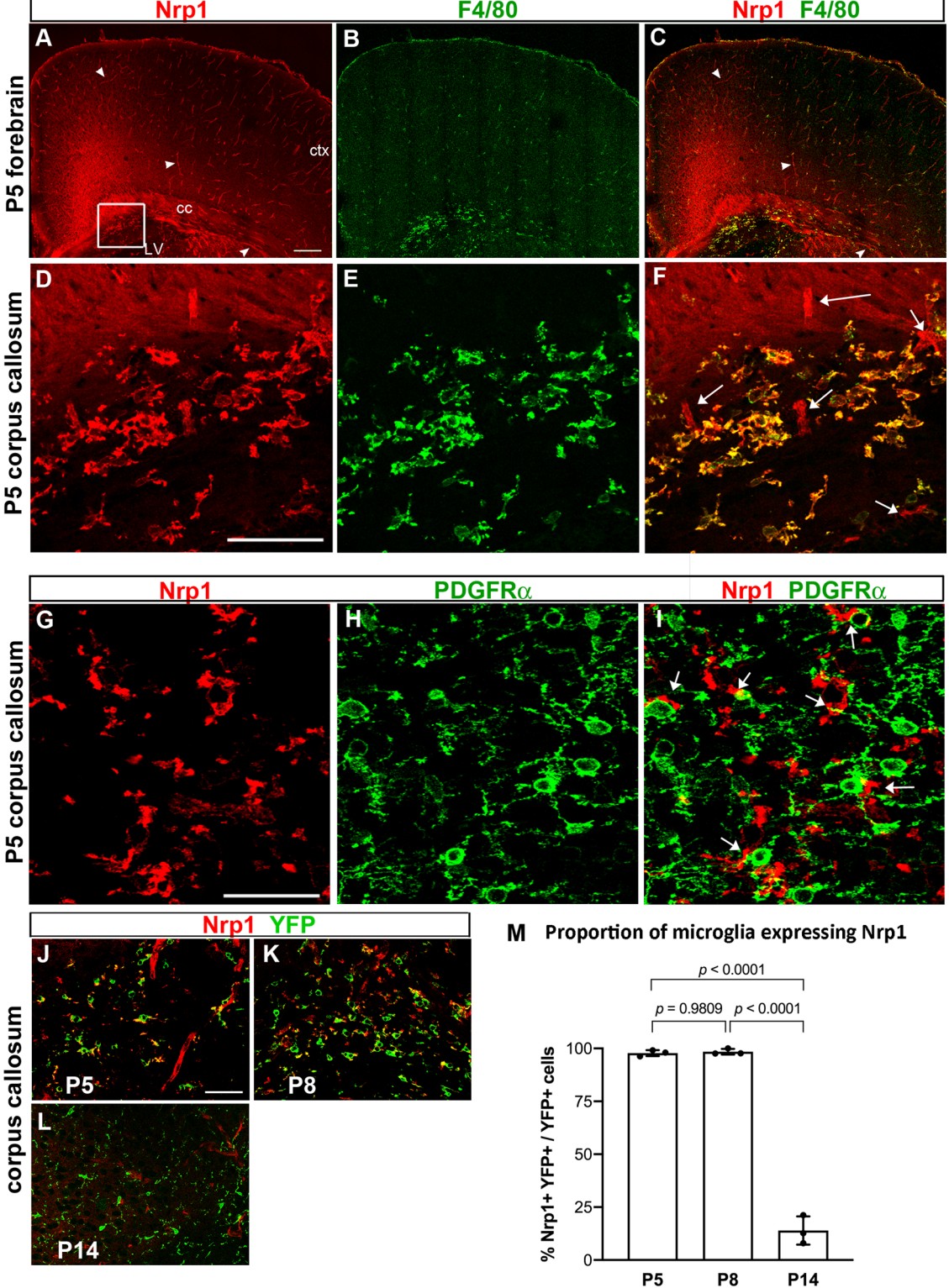

**Fig. 2 Nrp1 expression in the developing postnatal brain. A–C** Low-magnification view of a coronal forebrain section labeled for Nrp1 (red) and F4/80 (green). Arrowheads, examples of Nrp1+ blood vessels. Arrowheads, blood vessels. Boxed area with a cluster of Nrp1+ cells is shown in **D–F**. Scale, 100 µm. ctx, cortex; cc, corpus callosum; LV lateral ventricle. **D–F** High-magnification view showing Nrp1+ cells in the corpus callosum express F4/80. Arrows in **F** indicate blood vessels that are F4/80-negative. Scale, 50 µm. **G–I** Double labeling for Nrp1 (red) and PDGFRα in P5 corpus callosum. Nrp1 is not expressed on PDGFRα + OPCs, but Nrp1+ processes are found close to OPC cell bodies and processes (arrows in I). **J–L** Double labeling for Nrp1 (red) and YFP (green) in Cx3CR1^creERT2-ires-EYFP mice from P5 through P14. Scale = 50 µm. **M** The proportion of EYFP + microglia that expressed Nrp1 in the corpus callosum. One-way ANOVA, Tukey's multiple comparisons test, $n = 3$, $F(2, 6) = 438.8$. Source data are provided as a Source Data file.

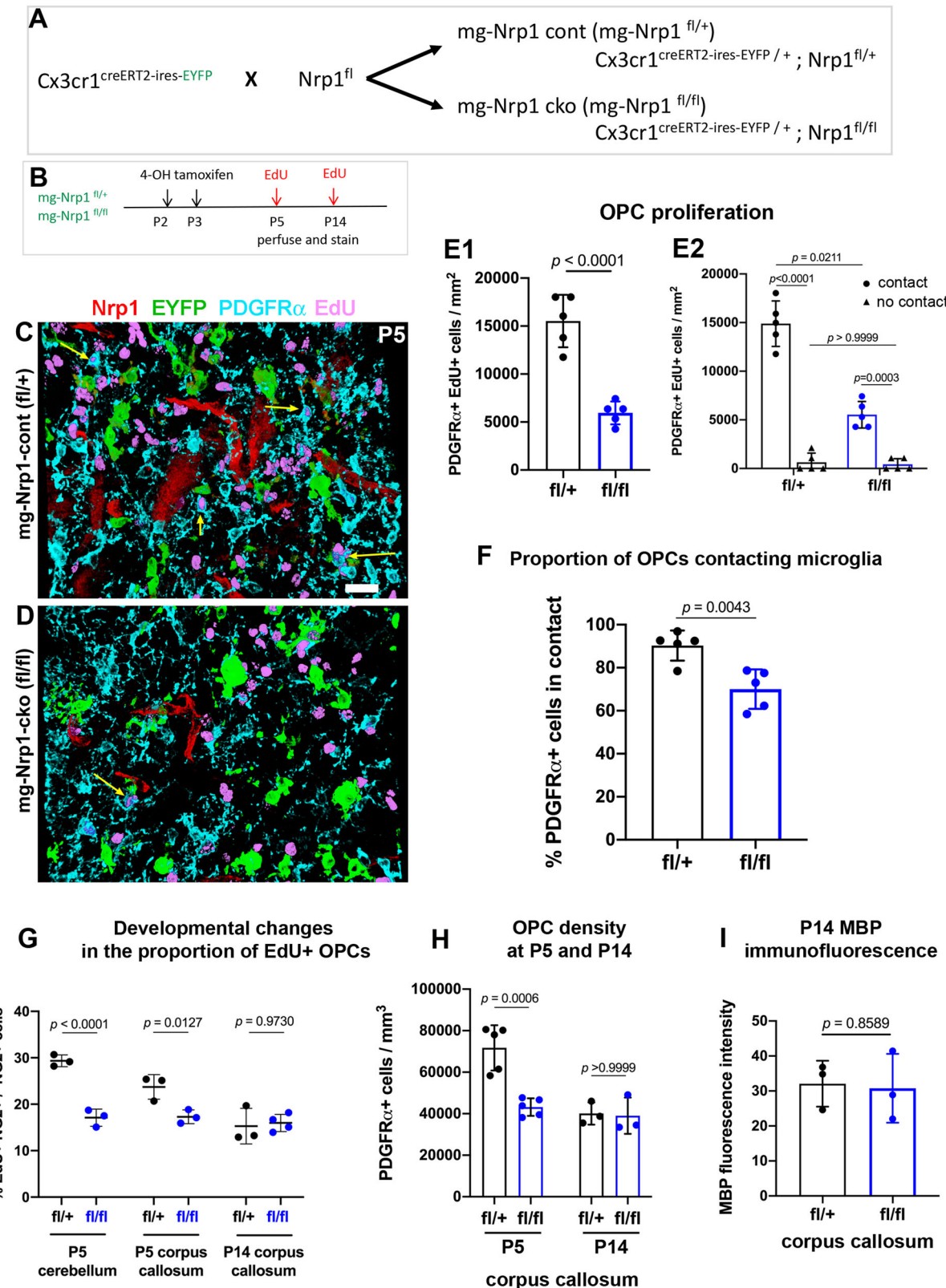

decrease in OPC proliferation in the corpus callosum of P5 mg-Nrp1-cko was largely due to the decrease in OPCs that were in contact with microglia. In mg-Nrp1-cko, the proportion of total PDGFRα+ OPCs that were contacting microglia, regardless of EdU incorporation, was 22% lower than that in mg-Nrp1-cont (Fig. 3F), suggesting that the presence of Nrp1 on microglia could be facilitating OPC contact.

We next examined whether the decrease in OPC proliferation in the white matter of mg-Nrp1-cko mice was restricted to the period in which Nrp1 was highly expressed on ameboid microglia. While the proportion of NG2 + OPCs that had incorporated EdU+ was significantly lower in the cerebellar white matter and corpus callosum of P5 mg-Nrp1-cko, the difference was no longer seen in P14 corpus callosum (Fig. 3G).

**Fig. 3 Effects of mg-Nrp1 cko on OPC proliferation in the white matter. A** Schematic of generating microglia-specific Nrp1 knock out (mg-Nrp1-cko) and the heterozygous control (mg-Nrp1-cont). **B** Schematic of tissue analysis at P5 and P14 after cre induction at P2-3. **C**, **D**. 3D images of P5 corpus callosum labeled for Nrp1 (red), EYFP (green), PDGFRα (light blue), and EdU (pink). Scale, 20 μm. Arrows, examples of EdU+ PDGFRα+ cells in contact with EYFP + microglia. **E** Quantification of proliferation of OPCs in mg-Nrp1-cont (black) and cko (blue). **E1** Total EdU+ OPC density in P5 corpus callosum. Unpaired Student's $t$-test, $n = 5$, $t = 7.171$, df = 8. **E2** The density of EdU+ OPCs in cont or cko with (circles) or without (triangles) contact with microglia. Two-way ANOVA, Sidak's multiple comparisons test, $F(1, 16) = 53.41$, $p < 0.0001$, $n = 5$. **F** Proportion of PDGFRα+ OPCs that were in contact with EYFP + microglia. Unpaired Student's $t$-test, $n = 5$, $t = 3.940$, df = 8. **G** Proportion of OPCs that were EdU+ in P5 cerebellum, P5 corpus callosum, and P14 corpus callosum. Two-way ANOVA, Sidak's multiple comparisons test. $F(1, 13) = 32.44$, $p < 0.0001$, $n = 3$–4. **H** Quantification of PDGFRα+ OPC density in P5 and P14 corpus callosum. Two-way ANOVA, Sidak's multiple comparisons test. $F(1, 12) = 19.07$, $p = 0.0009$, $n = 5$ (P5), $n = 3$ (P14). **I** Quantification of MBP immunofluorescence intensity in P14 corpus callosum of cont and cko mice. $n = 3$, $t = 0.1896$, df = 4. Black symbols, mg-Nrp1-cont; blue symbols, mg-Nrp1-cko in **E**–**I**. Source data are provided as a Source Data file.

Thus, the developmental age during which microglial Nrp1 deletion reduced OPC proliferation coincided with the temporal window during which Nrp1 was expressed on ameboid microglia in the corpus callosum (Fig. 2M, Fig. S2).

The density of total OPCs in the corpus callosum of mg-Nrp1-cko mice was 1.7-fold lower than that in control mice at P5, but by P14 the difference was no longer detected (Fig. 3H). TUNEL labeling did not reveal any TUNEL + cells in the corpus callosum of mg-Nrp1-cont or mg-Nrp1-cko at P5 (Fig. S4H–J), suggesting that microglial deletion of Nrp1 did not increase cell death. The transient reduction in OPC density did not lead to long-lasting changes in myelination, as judged by comparable levels of myelin basic protein (MBP) immunofluorescence in the corpus callosum in P14 mg-Nrp1-cont and cko mice (Fig. 3I and Fig. S4K–P).

**Microglia-specific Nrp1 deletion reduces PDGF AA-mediated OPC proliferation in slice culture.** To examine whether microglial deletion of Nrp1 compromised the proliferative response of OPCs to PDGF AA, we prepared slice cultures from P8 mg-Nrp1-cont and mg-Nrp1-cko mouse forebrains after 4-OHT administration at P2-3 and examined the proliferation of OPCs in gray and white matter in response to 50 ng/mL PDGF AA (Fig. S6A). There was no effect of Nrp1 deletion on EdU incorporation into gray matter OPCs. By contrast, EdU incorporation into OPCs in the corpus callosum of mg-Nrp1-cko slices was reduced to 57% of mg-Nrp1-cont.

To further examine whether Nrp1-deleted microglia directly affected OPC proliferation, we cocultured OPCs with perinatal microglia from mg-Nrp1-cont and mg-Nrp1-cko forebrain and assayed for OPC proliferation in the absence or presence of 15 ng/mL PDGF AA (Fig. S6B). OPC proliferation was significantly lower when cocultured with mg-Nrp1-cko microglia compared with mg-Nrp1-cont microglia. In the presence of a non-saturating concentration (15 ng/mL) of PDGF AA, the significant increase in OPC proliferation observed in OPCs cocultured with mg-Nrp1-cont microglia was not observed in OPCs cocultured with mg-Nrp1-cko microglia. Thus, the presence of Nrp1 on microglia plays a significant role in promoting OPC proliferation. The compromised OPC proliferation observed with cko microglia even in the absence of exogenous PDGF AA could reflect either PDGF AA secretion from microglia or OL lineage cells present in the culture[28] or other PDGF AA-independent effects of microglia on OPC proliferation.

**Microglial Nrp1 deletion compromises OPC expansion after demyelination.** Our immunolocalization studies revealed that Nrp1 was abundantly expressed on ameboid microglia that appeared transiently in the corpus callosum during the first postnatal week and was downregulated on microglia by P14 as they became more ramified. To determine whether Nrp1 expression could be upregulated in pathological conditions

that are known to increase OPC proliferation, we used an acute chemically induced demyelination model created by injecting α-lysophosphatidylcholine (LPC, lysolecithin) into the corpus callosum of 2- to 3-month-old mg-Nrp1-cont and mg-Nrp1-cko mice (Fig. 4A). Nrp1 was deleted in microglia by tamoxifen injection prior to induction of demyelination, with a deletion efficiency of 99.5% of YFP+ cells. We first examined whether Nrp1 was re-expressed on microglia after demyelination in mg-Nrp1-cont mice. Three days after PBS injection, EYFP+ cells surrounding the injection site had the morphology of resting ramified microglia and did not express Nrp1 (Fig. 4B). By contrast, 3 days after LPC injection (3 dpl), there was strong activation of microglia, and Nrp1 was robustly upregulated on the activated EYFP+ microglia/macrophages (Fig. 4C). At 7 dpl, EYFP+ activated microglia/macrophages were abundantly detected and expressed the lysosomal protein CD68 in the demyelinated lesion of both mg-Nrp1-cont and mg-Nrp1-cko corpus callosum (Fig. 4D, E). In mg-Nrp1-cont lesions, Nrp1 immunoreactivity was detected on the surface of CD68+ macrophages (Fig. 4D, arrows). This indicated that Nrp1 was highly upregulated on activated microglia/macrophages that appeared in the demyelinated corpus callosum. In the cko lesions, the majority of the CD68+ cells lacked Nrp1 (Fig. 4E, arrowheads). While Nrp1 was undetectable in 99.5% of YFP+ cells at 3 dpl, we detected a small number of CD68+ YFP- cells that expressed Nrp1 in the cko corpus callosum at 7 dpl (Fig. 4E, arrows), which were likely to have entered the lesion from Cx3cr1-negative precursors. The density of EYFP+ microglia/macrophages was similar in mg-Nrp1-cont and mg-Nrp1-cko mice at 3 dpl (Fig. 4F), indicating that the lack of Nrp1 on microglia in mg-Nrp1-cko lesions did not affect their infiltration or activation.

At 3 and 7 dpl, we pulse-labeled LPC-injected mice with EdU and examined the effect of microglial Nrp1 deletion on OPC proliferation in the lesion. EdU incorporation into EYFP-negative, NG2+ OPCs at the lesion site in mg-Nrp1-cko was 1.9-fold lower than that in mg-Nrp1-cont lesions at 3 dpl and 3.2-fold lower at 7 dpl (Fig. 4F–K). OPC proliferation was 2- to 3-fold higher at 3 dpl compared to 7 dpl for both genotypes. The time course of OPC proliferation is consistent with our previous report[12] of an early proliferative response in the region immediately surrounding the core demyelinated lesion. These observations indicate that loss of Nrp1 on activated microglia significantly compromised the early proliferative response[29] of OPCs to acute demyelination.

**Microglial Nrp1 deletion compromises OL regeneration and myelin repair.** To examine whether the reduced proliferation of OPCs in mg-Nrp1-cko mice affected subsequent events in myelin regeneration, we examined the lesion at 14 dpl. Quantification of OLs in and around the lesion at 14 dpl revealed that the density of CC1+ OLs in mg-Nrp1-cko lesion was reduced to 63% of that in

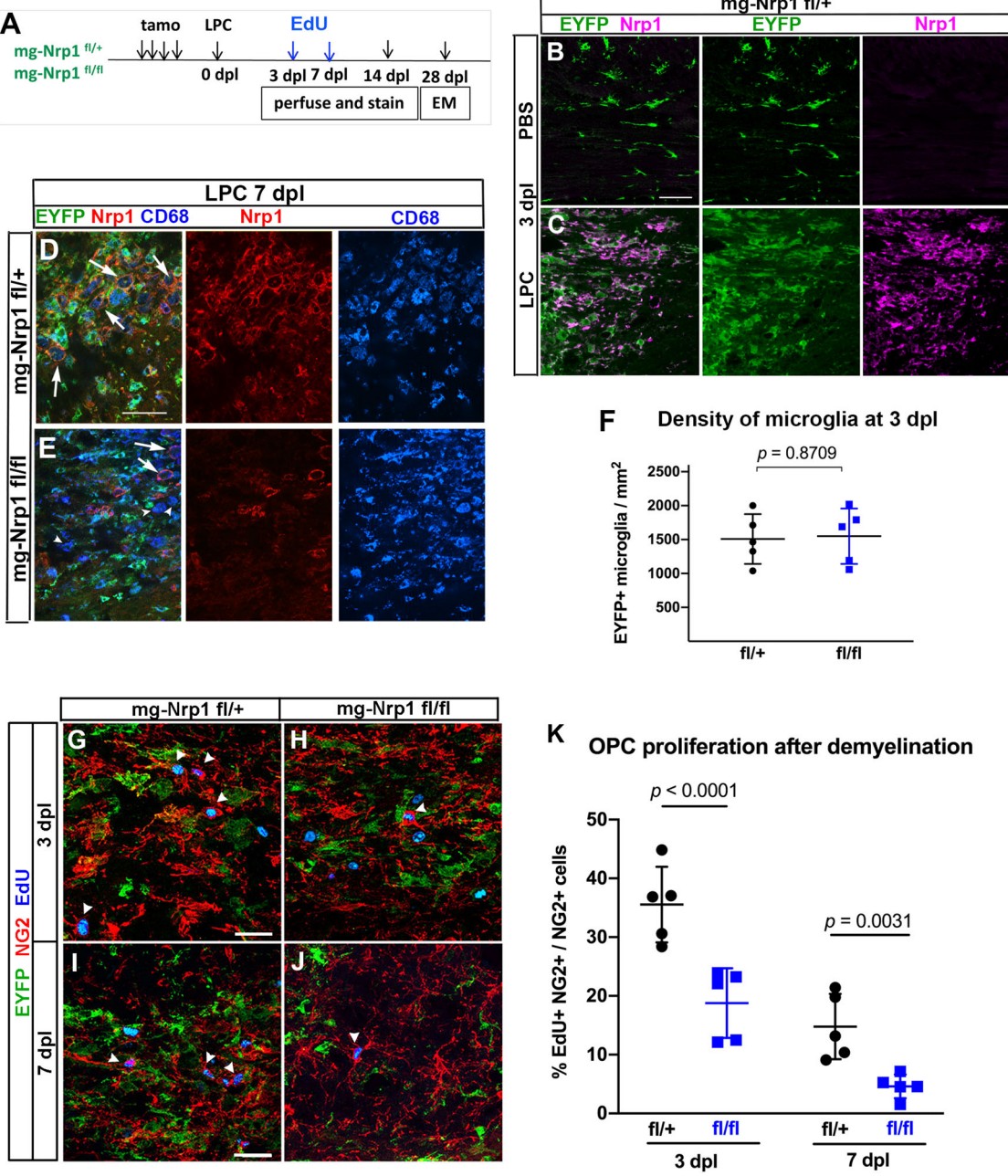

**Fig. 4 Effects of mg-Nrp1-cko on OPC proliferation and myelin repair after demyelination. A** Schematic of demyelination experiments. **B, C** Nrp1 (magenta) upregulation on EYFP+ microglia/macrophages 3 days after LPC injection but not after PBS injection. Scale, 50 μm. **D, E** Similar extent of CD68 (blue) expression and infiltration of activated microglia/macrophages into demyelinated lesion at 7 dpl in mg-Nrp1-cont (**D**) and mg-Nrp1-cko (**E**) mice. Red, Nrp1; green, EYFP. Arrows, Nrp1+ CD68+ macrophages; arrowheads, Nrp1-negative CD68+ cells. Scale, 50 μm. **F** Quantification of the density of EYFP+ microglia in the lesion at 3 dpl. Unpaired Student's $t$-test, $t = 0.1678$, $df = 8$, $n = 5$. **G–J** Labeling for EdU (blue), NG2 (red), and EYFP (green) in mg-Nrp1-cont and cko mice sacrificed at 3 dpl after EdU pulse labeling. Arrowheads, NG2+ EdU+ OPCs. Scale, 25 μm. **K** Quantification of proliferating OPCs at 3 and 7 dpl in cont (black) and cko (blue) mice showing significantly lower extent of EdU incorporation into OPCs in mg-Nrp1-cko mice. OPC proliferation was higher at 3 dpl compared to 7 dpl for both genotypes ($p = 0.0007$ for cont and $p = 0.0351$ for cko). Two-way ANOVA, Tukey's multiple comparisons test, $F_{(1, 16)} = 54.70$ for comparison between fl/+ and fl/fl, $F_{(1, 16)} = 32.52$ for comparison between 3 and 7 dpl, $n = 5$. Black circles, mg-Nrp1-cont; blue squares, mg-Nrp1-cko in **F** and **K**. Source data are provided as a Source Data file.

mg-Nrp1-cont lesion (Fig. 5A–C). To assess the extent of myelin repair, we first estimated the demyelinated area at 14 dpl by taking the area that exhibited reduced myelin basic protein (MBP) and elevated non-phosphorylated neurofilament immunoreactivity. The demyelinated area at 14 dpl was 1.74 times larger in the mg-Nrp1-cko mice compared with that in the mg-Nrp1-cont mice (Fig. 5D–F). Thus, the reduction of

oligodendrocyte density in the knockout animals correlated with the larger area containing demyelinated axons, suggesting that the repair process was impaired in mg-Nrp1-cko mice.

We next examined the impact of Nrp1 deletion from activated microglia on remyelination at 28 dpl. MBP immunofluorescence intensity in the lesioned corpus callosum of mg-Nrp1-cko mice was lower, at 61% of that in mg-Nrp1-cont mice (Fig. 5G–I).

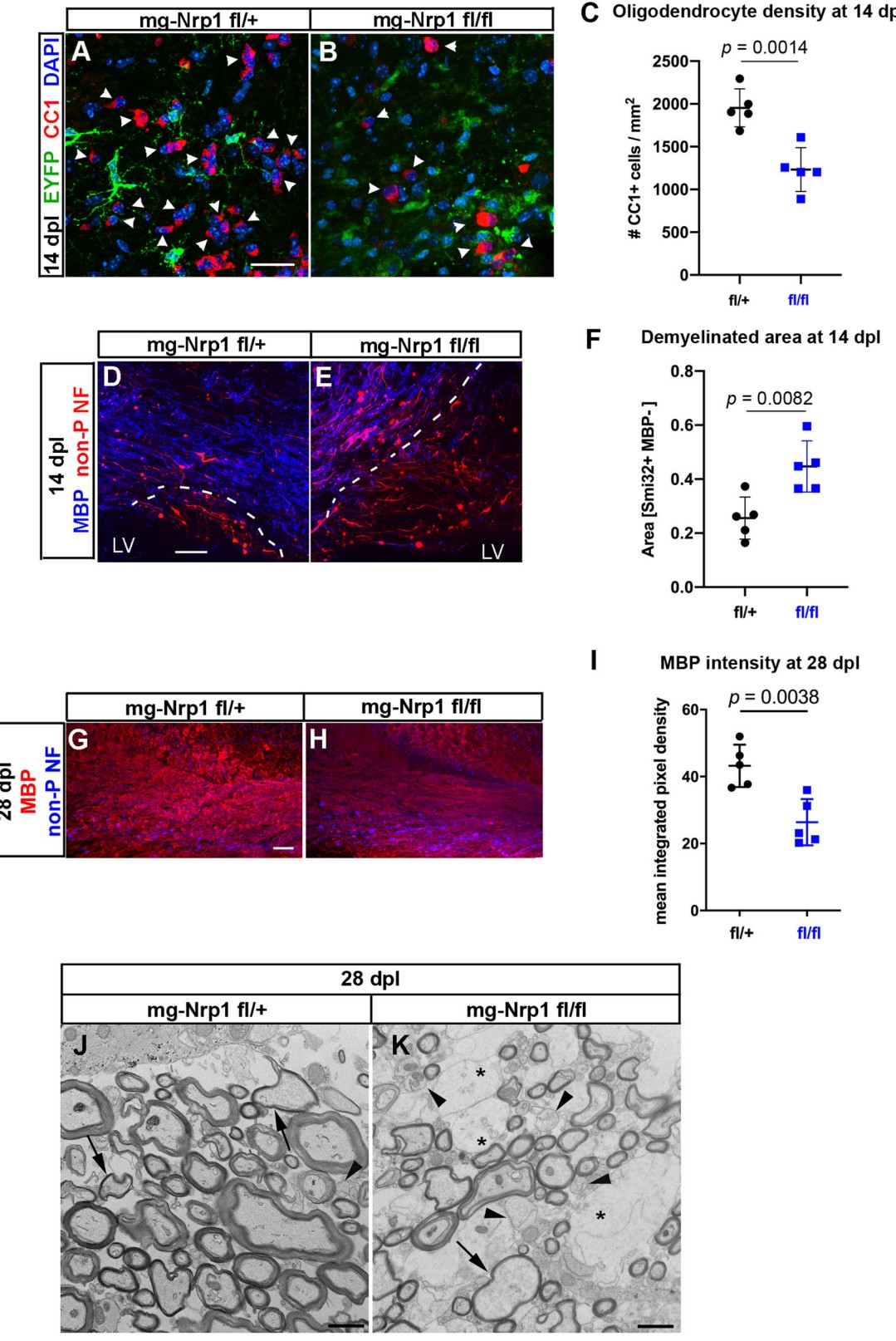

There was a higher level of non-phosphorylated neurofilament in the cko, which is consistent with prolonged impairment of myelin repair. To examine the extent of myelin repair in more detail, we performed ultrastructural analysis of the lesioned corpus callosum of mg-Nrp1-cont and mg-Nrp1-cko mice at 28 dpl (Fig. 5J–K and Fig. S7A–F) and searched for signs of myelin repair process. In mg-Nrp1-cont lesions, we saw that most axons were myelinated (Fig. 5J), and that some axons were surrounded by thin or less compacted myelin sheaths indicating remyelination (arrows in Fig. 5J), while very few axons were found

**Fig. 5 The extent of oligodendrocyte regeneration and myelin repair in mg-Nrp1-cko. A, B** Labeling mg-Nrp1-cont (L) and cko (M) for CC1 (red), DAPI (blue), and EYFP (green) at 14 dpl. Arrowheads, CC1+ cells. Scale, 25 μm. **C** Quantification of OL density in cont and cko lesions at 14 dpl. Student's *t*-test, unpaired, two-tailed, *n* = 5, *t* = 4.753, df = 8. **D, E** Labeling mg-Nrp1-cont (O) and cko (P) for MBP (blue) and non-phosphorylated neurofilaments (non-P-NF, red) using smi-32 antibody to identify axons that have not yet undergone complete remyelination. Dotted lines indicate the extent of partial or complete demyelination. Scale, 50 μm. LV, lateral ventricle. **F** Quantification of demyelinated area in cont and cko lesions at 14 dpl. Student's *t*-test, unpaired, two-tailed, *n* = 5, *t* = 3.490, df = 8. **G, H** Labeling for MBP (blue) and non-phosphorylated neurofilaments (non-P NF, red) in the lesioned corpus callosum of mg-Nrp1-cont (**G**) and cko (**H**) mice at 28 dpl. **I** Quantification of MBP immunofluorescence in the lesioned corpus callosum of mg-Nrp1-cont and cko mice at 28 dpl. Student's *t*-test, unpaired, two-tailed, *n* = 5, *t* = 4.031, df = 8. Black circles, mg-Nrp1-cont; blue squares, mg-Nrp1-cko in **C**, **F**, and **I**. **J, K** Electron microscopic images of cross-sections of LPC-lesioned corpus callosum at 28 dpl. **J** mg-Nrp1-cont (fl/+) showing sheaths of various thickness (arrows), but most axons appear fully myelinated. **K** mg-Nr1p-cko (fl/fl) showing more axons without myelin sheaths (arrowheads), axons with thin myelin (arrow), and abundant swollen glial cell processes (*). Scale bars 1 μm. Source data are provided as a Source Data file.

unmyelinated (arrowhead in Fig. 5J). By contrast, in the cko lesions (Fig. 5 K), there were more axons without myelin (arrowheads in Fig. 5K), and the tissue appeared more heterogeneous compared to the control and contained swollen glial processes (asterisks in Fig. 5K). We observed thin myelin and less compacted myelin sheaths (arrow in Fig. 5K), but overall, less myelin as compared to mg-Nrp1-cont (Fig. 5j). Some glial processes were present in the lesions of mg-Nrp-1-cont mice (Fig. S7A), but more glial cell processes as well as glial cells were present in mg-Nrp1-cko mice (Fig. S7D). We detected an oligodendrocyte in the process of ensheathing an axon (arrow in Fig. S7B) in the lesion of mg-Nrp1-cont mice, while the oligodendrocyte in mg-Nrp1-cko lesion was surrounded by several axons with un-compacted to more compacted myelin (Fig. S7E). Concomitantly, many axons displayed a glial ensheathment that was not yet compacted in the lesions of mg-Nrp1-cko mice (arrows in Fig. S7F), while this was less abundant in the mg-Nrp1-cont mice (arrow in Fig. S7C). These observations suggest that the remyelination process in mg-Nrp1-cko mice was not completely blocked but greatly delayed compared to the controls. Thus, failure to upregulate Nrp1 on activated microglia after acute demyelination significantly impaired the initial proliferative response of OPCs and compromised the timely production of new oligodendrocytes, causing a significant delay in the subsequent remyelination process.

**Exogenous Nrp1 augments PDGF-dependent OPC proliferation by augmenting PDGFRα phosphorylation on OPCs.** The above experiments revealed that Nrp1 on microglia is critical for PDGF AA-mediated proliferation of OPCs in the white matter and for the proliferative response of OPCs to acute demyelination. We next examined whether excess Nrp1 could augment the proliferative response of OPCs. Exogenous Nrp1 was added to slice cultures from P8 NG2cre;Z/EG mice in the form of soluble Nrp1-Fc fusion protein. Nrp1-Fc consisted of the extracellular domain of rat Nrp1, which had 98% amino acid identity with mouse Nrp1, fused to the Fc region of human immunoglobulin IgG1 in a homodimeric fusion protein construct. Addition of Nrp1-Fc to slice cultures in the presence of 50 ng/mL PDGF AA led to a significant increase in OPC proliferation in the cortex at 0.4 and 2 μg/mL of Nrp1-Fc (Fig. 6A–C, G). By contrast, proliferation of OPCs in the corpus callosum was unaffected by all concentrations of Nrp1-Fc tested (Fig. 6D–G). In the cortex, OPC proliferation reached 36% in the presence of 2 μg/mL Nrp1-Fc but remained lower than the 50% levels seen in white matter.

To determine whether the ability of exogenous Nrp1-Fc to augment PDGF-induced proliferation in gray matter was mediated by PDGFRα, we incubated slice cultures with function-blocking goat antibody to mouse PDGFRα in the presence of PDGF AA. In the absence of Nrp1-Fc, addition of 1 μg/mL of anti-PDGFRα antibody reduced OPC proliferation to 13.3% in gray matter and to

22.5% in white matter (Fig. 6H). In the presence of Nrp1-Fc, OPC proliferation in the cortex was increased to 35.8%, and this was reduced to 12.5% by anti-PDGFRα antibody (Fig. 6H). Similarly, addition of anti-PDGFRα antibody to Nrp1-Fc-treated cultures reduced OPC proliferation in white matter to 26.9% (Fig. 6H), similar to the level in the absence of Nrp1-Fc. These observations indicate that the increased OPC proliferation in gray matter elicited by Nrp1 was mediated through PDGFRα.

Since slice cultures contained different cell types, the above effects of Nrp1-Fc on OPCs could have been mediated by direct effects of Nrp1-Fc on OPCs or by indirectly altering signaling pathways in microglia or other cell types. To resolve this, we added Nrp1-Fc to purified dissociated OPC cultures and examined their proliferation in response to PDGF AA. OPCs were immunopanned from P2-4 mouse neocortex, and 2 μg/mL Nrp1-Fc was added together with 0 to 50 ng/mL PDGF AA (Fig. 7A–E). Immunopanning yielded an enriched population of OPCs containing <2% F4/80+ microglia. In the presence of PDGF AA concentrations of 4 μg/mL or lower, Nrp1-Fc had no effect on OPC proliferation, and fewer than 4% of Olig2+ OPCs were EdU+ (Fig. 7A, B, E). In the presence of 50 μg/mL PDGF AA, 54.6% of Olig2+ cells had incorporated EdU without Nrp1-Fc, and this was further increased to 75.9% by the addition of Nrp1-Fc. The most prominent effect of Nrp1-Fc was seen in the presence of 15 μg/mL of PDGF AA, increasing EdU incorporation 3.7-fold from 9% to 33.7% (Fig. 7C–E). These findings indicate that Nrp1 acted directly on OPCs to augment PDGF AA-induced proliferation, and it was most effective at suboptimum concentrations of PDGF AA.

Next, we determined whether Nrp1-Fc could augment PDGFRα activation on OPCs exposed to limited amounts of PDGF AA. PDGFRα activation was assessed by immunoblotting for tyrosine phosphorylation on PDGFRα after treating OPCs for 30 min at 37°C with 15 μg/mL PDGF AA with or without 2 μg/mL Nrp1-Fc. Immunoblots of protein extracts from the treated cells were incubated with rabbit and goat antibodies that recognized phosphorylated PDGFRα and total PDGFRα, respectively. We detected a significant increase in the level of phosphorylated PDGFRα in Nrp1-Fc-treated cells compared with those treated with PDGF AA alone (Fig. 7F, G). This indicated that Nrp1-Fc could augment the ability of PDGF AA to phosphorylate PDGFRα on OPCs.

To further examine the interaction between Nrp1-Fc and PDGFRα on OPCs, we performed co-clustering experiments by incubating dissociated OPCs with control-Fc or Nrp1-Fc for 30 min at 4°C or 36°C and stained for Fc and PDGFRα, as well as Olig2. After incubating OPCs with control-Fc consisting of human IgG Fc dimer at 4°C or 36°C or with Nrp1-Fc at 4°C (Fig. S8A–C), PDGFRα was found in small puncta along the processes and some at the soma (A, B), and there was little detectable distinct Fc immunoreactivity. Incubation of OPCs with Nrp1-Fc at 36°C caused greater aggregation of PDGFRα, and Fc immunoreactivity was also found co-clustered with many of the PDGFRα+ aggregates (Fig. S8D, arrows). These observations

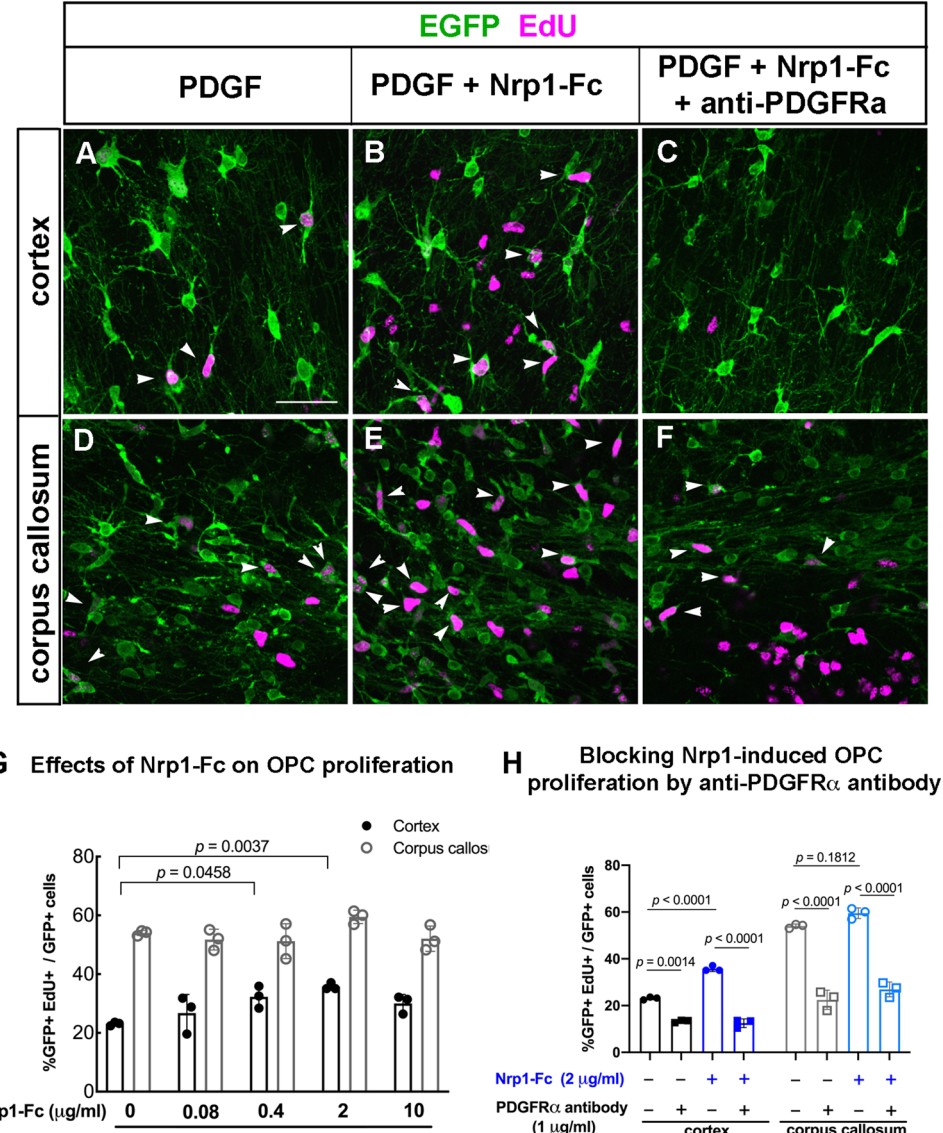

**Fig. 6 Effects of Nrp1-Fc on OPC proliferation in slice cultures from P8 NG2cre;Z/EG mice. A–F** Labeling for EGFP (green) and EdU (magenta) in the cortex (**A–C**) and corpus callosum (**D–F**) of slices treated with 50 ng/mL PDGF AA only (**A, D**), 50 ng/mL PDGF and 2 µg/mL Nrp1-Fc (**B, E**), or 50 ng/mL PDGF AA, 2 µg/mL Nrp1-Fc, and 1 µg/mL goat anti-mouse PDGFRα function-blocking antibody (**C, F**). Scale, 50 µm. **G** Dose-response of OPC proliferation in the cortex and corpus callosum to exogenous Nrp1-Fc in the presence of 50 ng/mL PDGF AA. Values are the percentages of EGFP+ cells that were EdU+. Two-way ANOVA, Tukey's multiple comparisons test, $n = 3$, $F_{(4, 20)} = 5.622$ for comparisons among different Nrp1-Fc concentrations and $F_{(4, 20)} = 317.6$ for gray versus white matter comparison. Black, cortex; gray, corpus callosum. **H** The effects of anti-PDGFRα blocking antibody on PDGF AA-mediated OPC proliferation in the cortex (gray bars) and corpus callosum (white bars) in the presence (checkered bars) or absence (solid bars) of exogenous Nrp1-Fc. All samples were treated with 50 ng/mL PDGF AA. Two-way ANOVA, Sidak's multiple comparisons test, $n = 4$, $F_{(3, 16)} = 266.6$. Circles, no blocking antibody; squares, PDGFRα blocking antibody. Black and gray, no Nrp1-Fc; blue, 2 µg/mL Nrp1-Fc. Source data are provided as a Source Data file.

further suggest that Nrp1-Fc binds to and causes functional clustering of PDGFRα.

## Discussion

We have shown that Nrp1 expressed by activated microglia plays a critical role in promoting OPC proliferation in early postnatal white matter tracts and after demyelination in the adult corpus callosum. Nrp1 was detected on the majority of ameboid microglia that appeared transiently in the early postnatal white matter, was not detected on resting ramified microglia, and was re-expressed on activated microglia/macrophages after

demyelination. During both development and after demyelination, Nrp1+ microglia were closely apposed to OPC processes. In slice culture, deletion of Nrp1 in microglia significantly reduced PDGF AA-induced OPC proliferation in white but not gray matter. In vivo, microglial Nrp1 deletion significantly reduced OPC proliferation, and the majority of OPCs that were proliferating in P5 corpus callosum were in contact with microglia. Microglial deletion of Nrp1 had no effect in regions where microglia were ramified and did not express Nrp1, such as the cortex and the mature corpus callosum. In adult corpus callosum, Nrp1 was robustly induced on activated microglia/macrophages after acute demyelination, and loss of Nrp1 in these cells

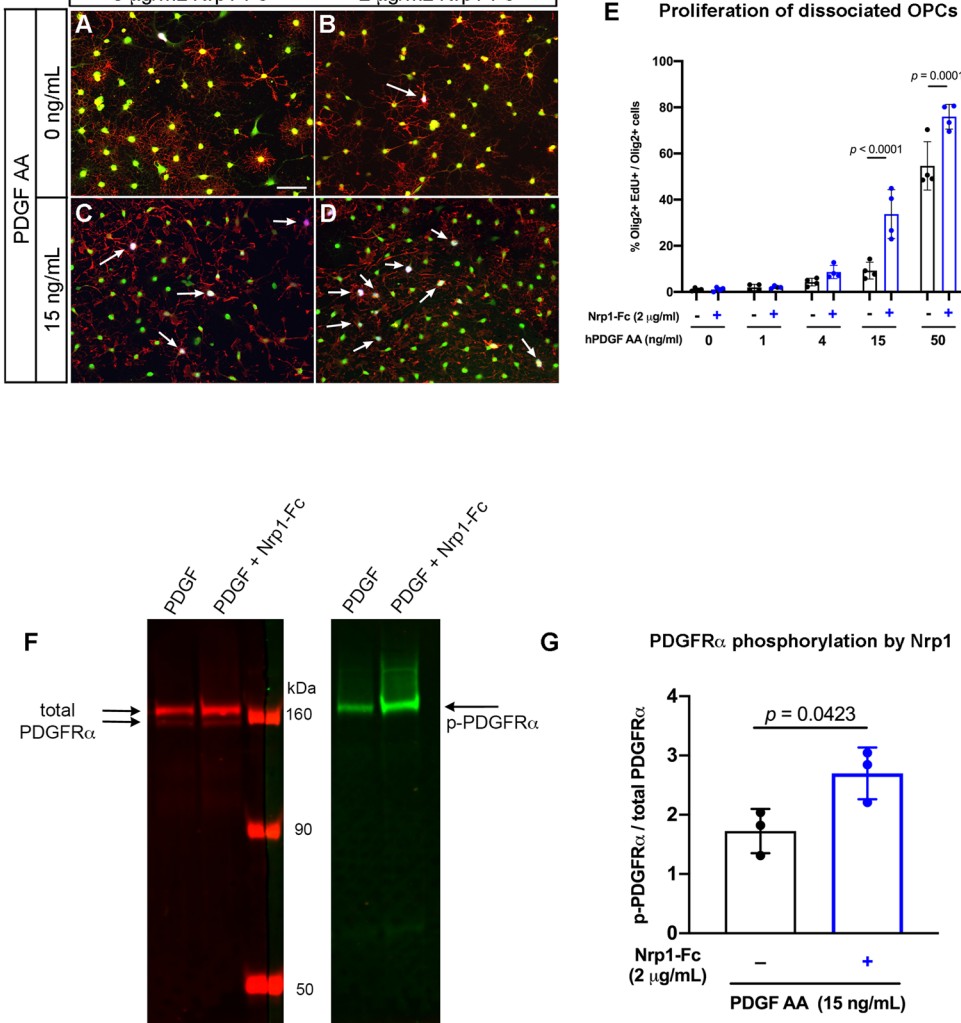

**Fig. 7 Effects of Nrp1-Fc on dissociated cultures of OPCs. A–D** Immunolabeling for Olig2 (green), NG2 (red), and EdU (gray) of cells grown in no PDGF AA (**A**, **B**) or 15 μg/mL PDGF AA in the absence (**A**, **C**) or presence of 2 μg/mL Nrp1-Fc (**B**, **D**). Scale in **A**, 50 μm. Arrows: EdU+ Olig2+NG2+ cells. **E** Quantification of the effects of combination of PDGF AA and Nrp1-Fc on OPC proliferation. Two-way ANOVA, Sidak's multiple comparisons test, $n = 4$, $F(1, 30) = 36.29$. **F** Immunoblots of OPCs treated with 15 ng/mL of PDGF AA alone or 15 ng/mL of PDGF AA and 2 μg/mL Nrp1-Fc immunostained with antibody to total PDGFRα (left, red) or phosphorylated PDGFRα (right, green). Arrows indicate the expected bands for PDGFRα and phosphorylated PDGFRα. **G** Quantification of the intensity of the phosphorylated PDGFRα band relative to total PDGFRα bands. **$p = 0.0081$, Student's $t$-test, paired, two-tailed $n = 3$, $t = 2.453$, df $= 2$. Black, absence of Nrp1-Fc; blue, presence of 2 μg/mL Nrp1-Fc. Source data are provided as a Source Data file.

significantly reduced OPC proliferation and subsequent oligo-dendrocyte differentiation and remyelination. Exogenous Nrp1 augmented PDGF AA-induced proliferation of cultured OPCs by increasing phosphorylation and clustering of PDGFRα. These findings support the model in which Nrp1 on microglia modulates PDGFRα-mediated proliferation of adjacent OPCs in trans (Fig. 8).

**Dynamic expression of Nrp1 on a subset of "activated" microglia in white matter.** CNS parenchymal microglia originate from the embryonic yolk sac and begin to colonize the brain after embryonic day 9.5 (E9.5)[30]. They exist in different inter-convertible functional states. In the normal mature brain, they exist as resting ramified microglia, or "homeostatic microglia", with highly branched processes[31,32] and are involved in a wide range of homeostatic regulation under physiological conditions[33,34], as well as in their well known immune function[35]. In response to a variety of insults, ramified microglia transform

into activated phagocytic microglia that become rounded in morphology and upregulate the lysosomal protein CD68[36]. Microglia also exist in the early postnatal white matter tracts as ameboid microglia with phagocytic activity before they transform into ramified microglia[37].

Nrp1 was expressed transiently on ameboid microglia in P5-8 corpus callosum and cerebellar white matter but not on ramified microglia in the cortex and became undetectable on microglia by P30. The developmental window during which Nrp1 was detected on ameboid microglia correlated with the period during which microglia-specific deletion of Nrp1 reduced OPC proliferation. Furthermore, this coincided with the period of rapid OPC proliferation and OL differentiation.

Resting ramified microglia in the mature CNS no longer expressed Nrp1. However, within three days following acute demyelination, activated CD68+ microglia/macrophages robustly upregulated Nrp1 on the cell surface. Deletion of Nrp1 from microglia significantly reduced OPC proliferation in the lesion

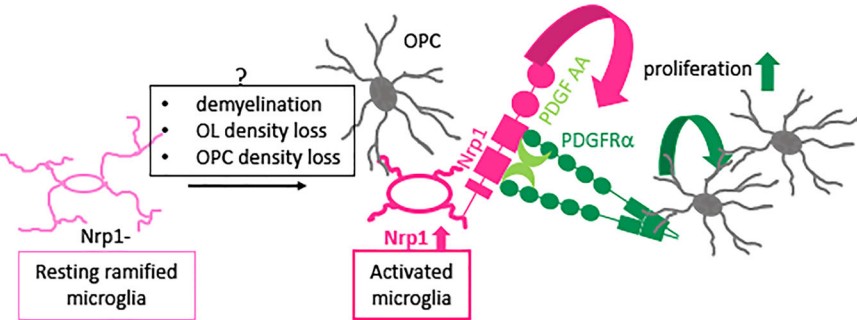

**Fig. 8 Proposed mechanism of action of microglial Nrp1 on OPC proliferation.** Resting ramified microglia (homeostatic microglia) that do not express Nrp1 (left, light pink) become activated and upregulate Nrp1. Nrp1 on activated microglia stimulates phosphorylation of PDGFRα on adjacent OPCs (gray) and promotes PDGF AA-dependent OPC proliferation. Question mark above the gray box indicates that the mechanism that causes Nrp1 upregulation on activated microglia is unknown.

during the first week after demyelination and subsequently led to reduced OL differentiation and myelination. Although the signals that dynamically upregulate Nrp1 on activated microglia in the developing and demyelinated white matter remain unknown, it is possible that phagocytosis of dead OL lineage cells by ameboid/activated microglia, a process which has been previously reported[37,38], could trigger a response that activates their Nrp1 expression.

*Transcriptomic similarities between early postnatal ameboid microglia and demyelination-induced activated microglia.* Recent single-cell transcriptomic studies have identified a transcriptionally unique subtype of microglia/monocytic cells in the mouse brain that are found in early postnatal white matter, which are referred to as axon tract-associated microglia (ATM)[39] or early postnatal proliferative region-associated microglia (PAM)[40], where oligodendrogliogenesis is actively occurring prior to myelination. Intriguingly, among microglia from five different ages and from demyelinated lesions, Nrp1 transcript counts were highest in those from P4/5 brain and LPC-induced white matter lesions[39] http://www.microgliasinglecell.com/), whereas Nrp1 expression was lower in the homeostatic microglia in the mature CNS. Thus, the two subpopulations of phagocytic microglia not only share their transcriptome but they also regulate OPC proliferation similarly by dynamically modulating Nrp1 expression on their surface.

**Beneficial effects of microglia on OL/myelin production.** Traditionally, microglia have been associated with inflammation and deleterious effects on OLs and myelin in demyelinating lesions of multiple sclerosis[41]. However, beneficial effects of microglia during myelin repair have also been identified[42]. Their dichotomous role appears to depend on their activation state[43,44]. During normal development, ameboid microglia/ATM/PAM in the early postnatal white matter tracts play a critical role in myelination. This is mediated in part by insulin-like growth factor-1 (IGF1)[45], which is highly expressed in ameboid microglia[39,40]. Pharmacological or genetic perturbation of the activation state of the ameboid microglia causes deficits in OL development and myelination[29,46,47]. However, the mechanisms by which microglia impart these positive effects on OL lineage cells have remained unclear.

Our studies have revealed that Nrp1, which is highly expressed on activated microglia in white matter, plays a critical role in promoting OPC proliferation during development and after demyelination. Developmental deletion of microglial Nrp1 did

not lead to permanent myelin defects, which could be due to compensation by proliferation and migration of microglia that had escaped Nrp1 deletion. Since we only deleted Nrp1 from microglia and not from vascular cells or axons, the partial inhibition of OPC proliferation in the mg-Nrp1-cko could be due to additional effects of Nrp1 on cell types other than microglia, as well as Nrp1-independent mechanisms that also promote developmental OPC proliferation.

Nrp1 loss from activated microglia that appeared in the demyelinated lesion led to a severe decrease in OPC proliferation followed by decreased OL regeneration and protracted remyelination. With compromised proliferative response of OPCs after acute demyelination, additional OPCs may have to be recruited from surrounding areas such as the subventricular zone or the gray matter, which could have contributed to the delayed remyelination. Interestingly, Nrp1 is highly expressed by microglia/macrophages associated with malignant glioma[48], and deletion of Nrp1 from microglia and macrophages slows glioma growth[49].

**Mechanism by which microglial Nrp1 promotes OPC proliferation.** Nrp1 is the axonal co-receptor for class III Semaphorins[21,50] and is critical for growth cone collapse induced by Sema3A[51]. Nrp1 was highly expressed on axons in E18.5 corpus callosum but the level had significantly dropped by P5. Axonal Nrp1 became undetectable in the corpus callosum after P14 and was not re-expressed after demyelination. Thus, axonal Nrp1 is not likely to be a major modulator of OPC proliferation. A related protein Nrp2 shares 44% amino acid identity with Nrp1 and binds primarily Sema3F. Semaphorins have been implicated in OPC migration during development[52,53] and after demyelination, acting via Nrp1 and 2 on OPCs[54,55]. However, we did not detect Nrp1 or 2 expression on OPCs, or on other cells in the corpus callosum, which was consistent with RNA-seq data for OPCs[28,56]. Neither did we detect Nrp2 on microglia, also consistent with the transcriptomic studies[28, 39,40]. Furthermore, the dramatic decrease in OPC proliferation in demyelinated lesions after Nrp1 deletion from activated microglia supports the view that microglial Nrp1 functions as a key inducer of OPC proliferation.

*Different saturation levels of Nrp1 in gray and white matter differentially affect OPC proliferation.* In slice cultures, blocking Nrp1 reduced PDGF AA-mediated OPC proliferation in the corpus callosum but not in the cortex. By contrast, addition of Nrp1-Fc augmented PDGF AA-mediated OPC proliferation in

gray but not white matter. Nrp1 may be present in saturating amounts in white matter, whereas in gray matter, the level of endogenous Nrp1 may be insufficient to adequately activate PDGFRα on OPCs. This is consistent with the immunohistochemical detection of higher levels of Nrp1 on activated microglia in white matter. This is also supported by the ability of Nrp1-Fc to augment proliferation and PDGFRα phosphorylation on dissociated OPCs in the presence of a suboptimum concentration of PDGF AA.

**Trans-activation of PDGFRα on OPCs by Nrp1 on adjacent microglia.** On mesenchymal stem cells, Nrp1 forms a complex with PDGFRα in the presence of PDGF and increases their migration and proliferation in response to PDGF AA[18]. In this case, both Nrp1 and PDGFRα are expressed in cis on the same cell, similar to the role of Nrp1 in $VEGF_{165}$-mediated VEGFR2 stimulation (reviewed in refs. [22,57]). By contrast, our findings support a model of trans-activation of PDGFRα on OPCs by Nrp1 on adjacent microglia (Fig. 8). Juxtacrine interaction between NRP1-expressing tumor cells and VEGFR2-endothelial cells can occur in vitro in the presence of $VEGF_{165}$[58]. In porcine aortic endothelial cells, presentation of human NRP1 in trans to VEGFR2 reduces internalization of VEGFR2 on the cell surface and causes delayed and sustained intracellular effects compared to cis-activation of VEGFR2 by NRP1[59]. Our observation that Nrp1-Fc co-clustered with PDGFRα on cultured OPCs and enhanced its phosphorylation is also consistent with the role of Nrp1 in stabilizing PDGFRα on the OPC surface. It is also possible that Nrp1 increases the availability of PDGF AA to PDGFRα by forming a ternary complex.

On other cell types, Nrp1 has been shown to be released by extracellular proteases[60,61]. However, if Nrp1 released from microglia were to affect OPC proliferation, the diffusion radius would have to be limited to the immediate pericellular environment, as the effects were confined to the corpus callosum and did not extend to the cortex. Furthermore, the immunofluorescence signal for Nrp1 was most prominent along the surface of microglia (Fig. 4D and Fig. S2). We previously reported a strikingly close apposition between an OPC and its adjacent microglia[62]. These observations, along with the close apposition between Nrp1+ microglia and OPCs observed here (Fig. 2), support a close contact-mediated interaction in trans. It is likely that Nrp1 signal constitutes a part of a larger constellation of microglia-to-OPC/OL signaling that maintains the homeostasis of OL and myelin density. Future studies may be directed to exploring the signaling mechanism of the cross-talk between Nrp1-expressing microglia and OPCs and how such interactions contribute to OL/myelin homeostasis.

## Methods

**Animals.** We have complied with all relevant ethical regulations for animal testing and research. All animal procedures received ethical approval by the Institutional Animal Care and Use Committee of the University of Connecticut. Mice were housed in a facility with 12:12 light:dark cycle, 50% humidity, and maintained at 73.5 °F. For slice cultures, we used NG2cre;Z/EG double transgenic mice[15], which were obtained by breeding NG2cre homozygous mice to Z/EG homozygous or heterozygous mice[63] (Jackson Laboratory stock no. 003920, Tg(CAG-Bgeo/GFP) 21Lbe/J; RRID:IMSR_JAX:003920). The NG2cre mice used in this study had been generated by injecting a BAC construct that contained NG2creER[TM], but one of the founders (NG2creERA) expressed cre independently of tamoxifen[64] and exhibited reporter expression in a temporal and spatial pattern that was identical to NG2creBAC;Z/EG mice previously described[65]. Since cre in the NG2creERA line was more stable than that in the original NG2creBAC mice described in ref. [65], the slice culture experiments described here were performed with NG2creERA;Z/EG mice, which we will refer to here as NG2cre;Z/EG mice. To delete Nrp1 in microglia, we crossed Nrp1[fl/+] or Nrp1[fl/fl] mice[26] (Jackson Laboratory stock no. 005247, B6.129(SJL)-Nrp1[tm2Ddg]/J; RRID:MGI:3528190) with Cx3cr1cre[ERT2-ires-EYFP] mice[27] (Jackson Laboratory stock no. 021160 B6.129P2(Cg)-Cx3cr1[tm2.1(cre/ERT2)Litt]/WganJ; RRID:MGI:5528845) to generate mg-Nrp1-cont or mg-Nrp1-cko mice,

respectively. Both males and females were used in all the experiments. Mouse genotyping primers are listed in Supplementary Table 1. To induce cre in mg-Nrp1-cont or mg-Nrp1-cko mice for developmental studies, we injected 4-hydroxytamoxifen (4-OHT, Sigma H7904, 100 µg/g) at P2 and P3 and used the mice at P8 for slice cultures or perfused them at P5, P8, P14, and P30 for histological analysis. To induce cre in weaned mice, 100 µg/g tamoxifen (Cayman Chemicals 13258 or Sigma T5648) was injected daily intraperitoneally for 4 consecutive days. To detect proliferating cells, 5-ethynyl-2'-deoxyuridine (EdU, 50 µg/g, Cayman Chemicals 20518) was injected intraperitoneally twice, 2 h apart, prior to sacrifice.

*Demyelinating lesion.* To induce demyelination, mice were anesthetized using isoflurane and their heads were placed in a stereotaxic apparatus. We injected 2 µl of 2.5% lysolecithin (LPC, α-lysophosphatidylcholine, Sigma L4129) dissolved in PBS into the corpus callosum of 9- to 12-week-old mg-Nrp1-cont and mg-Nrp1-cko mice, using the stereotaxic coordinates 0.3 mm anterior from the bregma, 1 mm lateral, and 1.9 mm from the surface of the skull[66]. The animals were perfused at 3, 7, 14, and 28 days post lesioning (dpl). To examine OPC proliferation, EdU was injected twice, 2 h apart, before sacrifice at 3 or 7 dpl.

**Tissue processing and immunohistochemistry.** Mice were perfused with 4% paraformaldehyde (PFA) containing 0.1 M L-lysine and 0.01 M sodium metaperiodate and postfixed in the same fixative for 2 h, after which the tissues were dissected and washed four times in 0.2 M sodium phosphate buffer, pH 7.4. The tissues were cryoprotected in 0.2 M sodium phosphate buffer containing 30% sucrose for at least 24 h, frozen in OCT compound (Tissue-Tek; Adwin Scientific 14-373-65), and 20-µm sections were cut on a Leica CM3050S cryostat. For free floating sections, 50-µm-thick coronal sections were cut with a Leica vibratome VT1000S directly after fixation and rinsing.

The antibodies are listed in Table 1. Rabbit antibody against the extracellular domain of rat PDGFα receptor was prepared by Dr. William Stallcup as follows. For generating the receptor fragment needed for immunization, a C-terminal His$_6$ sequence was added to a cDNA clone coding for the rat PDGFα receptor extracellular domain. This cDNA was ligated into the PCEP/4 vector and transfected into 293 EBNA cells, followed by hygromycin B selection to obtain positive colonies. After establishment of confluent monolayers of the transfected cells, the secreted his-tagged receptor fragment was purified from serum-free culture supernatant by chromatography on Ni$^{++}$-agarose (Qiagen). Authenticity of the purified material was confirmed by amino acid sequencing. Rabbit antisera produced against this receptor fragment were affinity-purified on a column constructed by coupling the purified receptor fragment to cyanogen bromide-activated Sepharose CL-4B (Pharmacia). Immunofluorescence tests using frozen sections of postnatal day 10 mouse brain confirmed that the affinity-purified antibody labeled OPCs in wild type specimens, but yielded no labeling in NG2 null specimens.

For immunolabeling, sections were rinsed in PBS, blocked in 5% normal goat serum (NGS) or 1% bovine serum albumin (BSA) containing 0.1% Triton X-100 in PBS for 1 h at room temperature, followed by incubation in the primary antibodies at 4°C overnight. The sections were rinsed three times in PBS and incubated in the secondary antibodies at room temperature for 1 h. Then the sections were rinsed and mounted with Vectashield with DAPI (Vector, H-1200). To detect EdU, immunolabeled sections were washed 3 times with PBS and incubated in the Click reaction mixture containing 150 mM NaCl, 100 mM TrisHCl pH 7.15, 4 mM CuSO$_4$.5H$_2$O (Sigma, C6283), 4 ng/mL Alexa Fluor-647-conjugated azide (ThermoFisher, A10277), and 100 mM sodium ascorbate (Sigma, A4034) at room temperature for 30 min. The sections were then washed three times in PBS, stained with 5 µg/mL Hoechst 33342 in PBS (InVitrogen Molecular Probes, LSH3570), and mounted in Vectashield without DAPI (Vector, H-1000). To examine cell death, we applied ApopTag Red In Situ Apoptosis Detection Kit (Millipore, S7165) according to the manufacturer's recommended procedure, after treating the tissues with pre-cooled ethanol:acetic acid (2:1) for 5 min.

**Transmission electron microscopy.** After perfusion and post-fixation of LPC-injected mice at day 28 dpl, brains were washed in PBS. One-hundred micrometer thick coronal slices were obtained by slicing the brains with a Leica vibratome VT1000S. Slices containing the LPC-induced lesion were selected and fixed again in 4% PFA and 2% glutaraldehyde in 0.1 M sodium phosphate buffer for 2 h. The corpus callosum within the region of interest was dissected and processed for resin embedding. Specimens were further fixed with 1% Osmium tetroxide at room temperature for 2 h. Tissue was dehydrated through a series of graded ethanol, during which 1.5% uranyl acetate was included. 100% ethanol was gradually mixed with an increasing portion of propylene oxide and finally incubated in propylene oxide. Subsequently, infiltration with Spurr's resin was carried out with an increasing portion of resin diluted with propylene oxide. Infiltration was completed with 100% epon resin. Tissue was embedded in flat double end molds and polymerized at 70°C for 48 h. Semithin sections were cut with a Diatome[TM] diamond knife on a Leica Ultracut UCT microtome and stained with methylene blue/azure blue. The region of interest was selected with upright Leica DMR and Zeiss AxiovertM200 inverted microscopes, and ultrathin sections were cut with an ultra 45° Diatome TM diamond knife. Sections were collected on 100 mesh copper grids. Images were obtained using a bright field FEI Tecnai Biotwin G2 Spirit

**Table 1 Primary and secondary santibodies used.**

| Antibody | Source | Dilution | Cat no. | RRID |
|---|---|---|---|---|
| Primary antibodies | | | | |
| Mouse anti-CC1 | Millipore | 1:200 | OP80 | RRID:AB_2057371 |
| Rat anti-CD140a, clone APA5 | BD Sciences | | 558774 | RRID:AB_397117 |
| Rat anti-CD68 | Biolegend | 1:1000 | 137001 | RRID:AB_2044003 |
| Rat anti-F4/80 | Bio-Rad | 1:200 | MCA497R | RRID:AB_323279 |
| Chicken anti-GFP | Aves Laboratories | 1:500 | GFP-1010 | RRID:AB_2307313 |
| Mouse anti- glyceraldehyde-3-phosphodehydrogenase (GAPDH) | Millipore | 1:2000 | MAB374 | RRID:AB_2107445 |
| Goat anti-Iba1 | Invitrogen | 1:500 | PA5-18039 | RRID:AB_10982846 |
| Rabbit anti-laminin | Sigma | 1:1000 | L8271 | RRID:AB_477162 |
| Mouse anti-myelin basic protein (MBP, SMI99) | Biolegend | 1:2000 | 808403 | RRID:AB_2734562 |
| Mouse anti-Neurofilament H (non-phosphorylated, SMI32; 1:3000) | Biolegend | 1:3000 | 801701 | RRID:AB_2564642 |
| Rabbit anti-NG2 | Millipore | 1:500 | AB5320 | RRID:AB_11213678 |
| Goat anti-Nrp1 | R&D Systems | 1:200 | AF566 | RRID:AB_355445 |
| Mouse anti-Olig2 | Millipore | 1:500 | MABN50 | RRID:AB_10807410 |
| Goat anti-PDGFRα | R&D Systems | 1:1000 | AF1062 | RRID:AB_2236897 |
| Rabbit anti-PDGFRα | Dr. William Stallcup[a] | 1:1000 | a | RRID_AB_2315173 |
| Rabbit anti-Phospho-Pdgfra | Cell Signaling Technology | 1:1000 | 3170 | RRID:AB_2162348 |
| Secondary antibodies | | | | |
| Alexa 488-donkey anti-chicken IgY | Jackson ImmunoResearch | 1:1000 | 703-545-155 | RRID:AB_2340375 |
| Cy3-donkey anti-mouse IgG (H + L) | Jackson ImmunoResearch | 1:500 | 715-165-150 | RRID:AB_2340813 |
| Alexa 647-donkey anti-mouse IgG (H + L) | Jackson ImmunoResearch | 1:200 | 715-605-150 | RRID:AB_2340862 |
| Cy3-goat anti-mouse IgG 2a | Jackson ImmunoResearch | 1:500 | 115-165-206 | RRID:AB_2338695 |
| Alexa 647-goat anti-mouse IgG 2a | Jackson ImmunoResearch | 1:200 | 115-605-206 | RRID:AB_2338917 |
| Cy3-goat anti-mouse IgG 2b | Jackson ImmunoResearch | 1:500 | 115-165-207 | RRID:AB_2338696 |
| Alexa 647-goat anti-Mouse IgG 2b | Jackson ImmunoResearch | 1:200 | 115-605-207 | RRID:AB_2338918 |
| Alexa 488-donkey anti-rat IgG (H + L) | Jackson ImmunoResearch | 1:1000 | 712-545-150 | RRID:AB_2340683 |
| Alexa 488-chicken anti-rat IgG (H + L) | Thermo Fisher Scientific | 1:1000 | A-21470 | RRID:AB_2535873 |
| Cy3-bovine anti-goat IgG (H + L) | Jackson ImmunoResearch | 1:500 | 805-165-180 | RRID:AB_2340880 |
| Cy3-donkey anti-goat IgG (H + L) | Jackson ImmunoResearch | 1:500 | 705-165-147 | RRID:AB_2307351 |
| Alexa 647-donkey anti-goat IgG (H + L) | Jackson ImmunoResearch | 1:200 | 705-165-147 | RRID:AB_2307351 |
| Brilliant violet 421-donkey anti-rabbit IgG (H + L) | Jackson ImmunoResearch | 1:200 | 711-675-152 | RRID:AB_2651108 |
| Cy3-donkey anti-rabbit IgG (H + L) | Jackson ImmunoResearch | 1:500 | 711-165-152 | RRID:AB_2307443 |
| Alexa488-goat anti-human IgG (H + L) | Thermo Fisher Scientific | 1:1000 | A-11013 | RRID:AB_2534080 |
| IRDye 800CW donkey anti-rabbit IgG | LI-COR | 1:15,000 | 926-32213 | RRID:AB_621848 |
| IRDye 680RD donkey anti-goat IgG | LI-COR | 1:5000 | 926-68074 | RRID:AB_10956736 |

[a]W.B. Stallcup, Sanford-Burnham Medical Research Institute Cat# PDGFRalpha, RRID:AB_2315173s.

(Netherlands) transmission electron microscope operated at an accelerating voltage of 80 kV and equipped with an AMT 2k (4 megapixel) XR40 CCD camera.

**Slice cultures.** Slice cultures from the forebrain and cerebellum were prepared from P8 NG2cre;Z/EG mice[15,67] or from P8 mg-Nrp1-cont or cko mice after 4-OHT injection at P2-3 in vivo. Forebrains were chopped into 300-μm thick slices and transferred to ice-cold dissection medium consisting of 124 mM NaCl, 2.004 mM KCl, 1.25 mM KH2PO4, 4.004 mM MgSO4 (anhydrous), 2.0 mM CaCl2·2H2O, 26 mM NaHCO3, 10 mM D-(+)-glucose, 2 mM ascorbic acid and 0.075 mM adenosine dissolved in water. Individual slices were separated and placed on Millicell culture inserts and maintained at air-liquid interface in a humidified incubator at 37 °C with 5% CO2 in slice media consisting of 50% Minimal Essential Medium with Earle's Salts, 25 mM HEPES buffer, pH 7.22, 25% HBSS without CaCl2, MgCl2, or MgSO4, 25% horse serum, 0.4 mM ascorbic acid, 1 mM L-glutamine, and 1 mg/L insulin. Culture medium was changed 24 h after dissection and every other day thereafter for 9 days. On day 7, PDGF AA (R&D Systems, 221-AA) was added to the cultures and the slices were incubated for 48 h. During the last 5 h of incubation, 10 μM EdU was added to the slice medium. To block Nrp1 or PDGFRα function, slices were incubated in different concentrations of goat anti-mouse Nrp1 antibody (R&D Systems, AF566) or 1 μg/mL of goat anti-mouse PDGFRα antibody (R&D Systems, AF1062) during the last 48 h of incubation (on day 7). Nrp1-Fc fusion protein containing the extracellular domain of rat Nrp1 (amino acid 22-854 minus 811-828; R&D Systems, 566-NNS) or the control human Fc dimer (R&D Systems, 100-HG) was added to slice cultures at the indicated concentrations for the last 48 h of incubation. At the end of the incubation, slices were fixed with 4% paraformaldehyde and processed for immunofluorescence labeling and EdU detection.

**Dissociated OPC cultures.** OPCs from the cerebral cortex of P4-5 wild type CD1 mice were immunopanned for PDGFRα using CD140a rat anti-mouse PDGFRα antibody[68]. The brains were scooped out from the opened skull and diced in an ice-cold D-PBS without Ca²⁺/Mg²⁺(Invitrogen) in the dish. The minced tissues were incubated in an oxygenated papain solution containing 200 units of papain (Worthington Biochemical, LS003126), 2 mg of L-cysteine (Sigma-Aldrich, C8786) and DNase I (Worthington Biochemical, L002007) on a 37 °C heating block for 30–45 min. Following tissue digestion, papain was inactivated using a low-ovo solution containing BSA (Sigma-Aldrich, A8806), D-PBS (Invitrogen), Trypsin inhibitor (Worthington Biochemical, LS003086). The brain tissues were dissociated by gently pipetting in low-ovo solution. Following centrifugation, the cell pellet was suspended in panning buffer consisting of 0.2% BSA and 2 mg/mL insulin in DPBS and immunopanned in 10-cm petri dishes precoated with CD140a rat anti-mouse PDGFRα antibodies for 45 min. Following 6-8 times wash with D-PBS, the adhered OPCs were trypsinized for 6–7 min, and trypsin was inactivated using 30% fetal bovine serum (FBS). Following centrifugation, the OPCs were resuspend in DMEM-Sato medium containing Dulbecco's Modified Eagle's Medium (Invitrogen), 2 mM L-glutamine, SATO supplements (1 μg/mL transferrin, 1 μg/mL BSA, 0.16 μg/mL putrescine, 0.6 ng/mL progesterone, 0.4 ng/mL sodium selenite), penicillin/streptomycin, 1 mM sodium pyruvate, 5 μg/mL insulin, 5 μg/mL N-acetyl-L-cysteine (Sigma-Aldrich, A8199), 1X Trace Elements B (Cellgro-Corning, Fisher Scientific MT99175Cl),10 ng/mL d-Biotin (Sigma-Aldrich, B4639) and 1X B-27 (Thermo Fisher, 17504044). Purified OPCs were plated on glass coverslips coated with poly-D-lysine (Sigma, P7405, 10 μg/mL) at a density of 40,000 cells/well in 24-well plates for EdU incorporation assays or on 35-mm tissue culture dishes coated with poly-L-lysine (Sigma, P1524; 30 μg/mL) at a density 10 × 10⁶ cells/dish for immunoblotting. For EdU detection, the cells on coverslips were initially incubated in DMEM-Sato medium and PDGF AA for 2 days, and 10 μM EdU was added to the cells for 6 h prior to staining for Olig2, NG2, and EdU. For detecting PDGFRα phosphorylation by immunoblotting, immunopurified OPCs were incubated with DMEM-Sato medium without PDGFAA for 12 h. Then, the medium was replaced with new medium containing 2 μg/mL Nrp1-Fc or control Fc protein and 15 ng/mL PDGFAA and incubated at 37 °C for 30 min, after which the cells were promptly chilled on ice for protein extraction.

For Nrp1-Fc-PDGFRα capping experiments, immunopanned OPCs from P4-5 CD1 mice were plated on PDL coated coverslips at a density of 40,000–50,000 cells/well in 24-well plates and incubated in DMEM-Sato medium overnight. Then, 2

µg/mL control human IgG1-Fc (cont-Fc) or Nrp1-Fc was added in the presence of 15 ng/mL PDGF AA and incubated at 4 °C or 36 °C for 30 min. The coverslips were washed with cold PBS, fixed with 4% PFA, and then immunolabeled with mouse anti-Olig2 (Millipore MABN50, 1:500) and rabbit anti-PDGFRα antibodies (obtained from Dr. William Stallcup, Sanford Burnham Institute, La Jolla, CA; 1:1,000 dilution). Following washes, cells were incubated in Alexa488-goat anti-human Fc, Cy3-bovine anti-rabbit, and Alexa647-donkey anti-mouse antibodies.

For OPC-microglia cocultures, we first injected 4-OHT into mg-Nrp1-cont and mg-Nrp1-cko mice from P2-3. On P5, microglia were isolated by immunopanning using rat anti-mouse F4/80 antibody (Bio-Rad). Purified OPCs from CD1 pups and microglia were plated together on glass coverslips coated with poly-D-lysine (Sigma, P7405, 10 µg/mL) at a density of 40,000–50,000 cells/well in 24-well plates and incubated overnight in DMEM-Sato medium with or without 15 ng/mL PDGF AA for 2 days. Ten micromolar EdU was added to the cells 6 h prior to fixation. Fixed cells were stained for Olig2 and EdU.

**Immunoblotting**. Purified immunopanned OPCs were rinsed with chilled PBS containing 1x PhosSTOP (Sigma-Aldrich-Roche, 4906845001) and 1x halt protease inhibitor cocktail (ThermoFisher, 87786) and harvested in 1 mL of the same buffer. The cell pellet was lysed by homogenizing in RIPA Buffer (ThermoFisher) supplemented with PhosSTOP and protease inhibitors on ice for 30 min and then centrifuged to remove insoluble matter. The cleared lysate was frozen in liquid nitrogen and stored at −80 °C. Protein concentration was measured using DC protein assay (Bio-Rad). Samples were denatured with 4X Bolt LDS sample buffer and 10X sample reducing agent (ThermoFisher B0007 and B0004, respectively) and heated at 70 °C for 10 min. Samples (30 µg/lane) and prestained molecular weight standards (LI-COR, 928–7000) were electrophoresed through 8% Bolt Bis-Tris plus polyacrylamide gels (ThermoFisher) in Bolt MOPS SDS running buffer (ThermoFisher). Proteins were transferred to Immobilon FL membranes (EMD-Millipore) using a transfer buffer containing 1.25 mM bicine, 1.25 mM bis-Tris, and 0.05 mM EDTA. Blots were blocked in Odyssey TBS (Tris-buffered saline) blocking buffer (LI-COR) at room temperature for 1 h. Blots were incubated at 4 °C overnight in primary antibodies diluted in TBS blocking buffer containing 0.1% Tween 20. The primary antibodies were rabbit anti-Phospho-PDGFRα (Cell signaling) and goat anti-mouse PDGFRα that recognizes both phosphorylated and non-phosphorylated forms (R&D Systems). Blots were washed with TBST buffer containing 137 mM NaCl, 20 mM TrisHCl, pH 7.6, and 0.1% Tween 20) four times, 10 min each, and incubated with IRDye 800RD donkey anti-rabbit or IRDye 680RD donkey anti-goat secondary antibodies diluted in Odyssey blocking buffer with 0.1% Tween 20 and 0.01%SDS. Blots were washed with TBST and imaged on a LI-COR Odyssey Imager. To quantify protein expression, the density of protein bands was determined using Image Studio Lite (LI-COR).

**Fluorescence microscopy and quantification**. Images of fluorescently labeled tissue sections, slice cultures, and dissociated cells were acquired using Leica SP8 confocal microscope (Advanced Light Microscopy Facility) or Zeiss Axiovert 200 M with apotome. Images were analyzed with the Leica LAS X, Zeiss Axiovision, or ImageJ. For quantification, images were captured from several random fields of defined area within the cortex, corpus callosum, or the cerebellar white matter, based on DAPI label and blind to the experimental labels, and immunolabeled cells in each field were counted.

To quantify the proliferative OPCs that were in contact with microglia, 0.35-µm z-stack images were acquired from 20-µm-thick sections using a 40X objective on a Leica SP8 confocal microscope at 1024 × 1024 pixel image size and 700 Hz scan speed. Three-dimensional images were reconstructed from the z-stacks using the 3D module of the Leica LAS X software. Each PDGFRα + OPC was marked and first examined for contact with Nrp1+ microglia by rotating the plane in all directions. Then, the EdU channel was revealed, and the OPC was scored for whether it had EdU label.

For quantification of the degree of myelination, sections from P14 corpus callosum or demyelinated corpus callosum at 28 dpl were immunolabeled for MBP, and MBP immunofluorescence intensity was quantified on images scanned on the Leica SP8 confocal microscope. Fiji (ImageJ, version 1.53a) was used to obtain the average integrated pixel density of gray values over a defined region of interest. Background values were obtained from areas devoid of signal.

For dissociated OPCs, images were taken from randomly selected fields from each coverslip based on DAPI label and the labeled cells were quantified. Adobe Photoshop was used to generate image panels for the figures. Quantification data are represented as means ± standard deviations. Statistical analyses were performed using GraphPad Prism v8 and v9. The specific statistical method used to evaluate each dataset is indicated in the figure legends.

**Reporting summary**. Further information on research design is available in the Nature Research Reporting Summary linked to this article.

## Data availability
The datasets generated and analyzed in this study are available from the corresponding author upon reasonable request. Source data are provided with this paper.

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

## Acknowledgements

We thank Youfen Sun for maintaining the mouse colony. This study was supported by NIH R01 NS073425 (A.N.) and funding from the European Union's Framework Program for Research and Innovation Horizon 2020(2014-2020) under the Marie Sklodowska-Curie Grant Agreement No. 845336 (F.P.). We thank Dr. Chris O'Connell, Director of Advanced Light Microscopy Core, for his assistance with the Leica SP8 confocal microscope, which was purchased with funds from an NIH Instrumentation Grant S10 OD016435 (PI: Akiko Nishiyama). Electron microscopy was performed at the Biosciences Electron Microscopy Facility of the University of Connecticut under the guidance of Director Dr. Linnaea Ostroff and with the excellent technical support of Drs. Maritza Abril and Xuanhao Sun. We thank Dr. Timothy Moore, Director of the Statistical Consulting Services, for his assistance with statistical analyses and graphing of the data.

## Author contributions

A.S., A.R., F.P., and A.N. designed the study. A.S., A.R., F.P., and W.M.W. generated data. A.S., F.P., W.M.W., and A.N. interpreted the data. A.S., F.P., and A.N. wrote the manuscript, and A.S., F.P., W.M.W., and A.N. edited the manuscript.

## Competing interests

The authors declare no competing interests.
