## [Peer Review File · Nature Communications]

Reviewers' Comments:

Reviewer #1:

Remarks to the Author:

This study by Sherafat et al. provides evidence that microglial neuropilin-1 (NRP1) signaling enhances PDGF AA-mediated proliferation of oligodendrocyte precursor cells (OPCs) during development and after lysolecithin-induced demyelination. While the effects in development appear to be transient and relatively mild, during demyelination were more robust with effects on generation of new oligodendrocytes and remyelination. Mechanisms by which microglia regulate other glial cells is an understudied, but potentially very important, area of research. These findings define novel signaling between microglia and OPCs to modulate OPC proliferation, response to injury, and remyelination. Enthusiasm is generally high, but there are some concerns that require attention outlined below.

Major points:

- To support the conclusions, the manuscript relies on the selective ablation of NRP1 in microglia. However, the data validating the microglia-specific NRP1 ablation are limited and are not quantified. Given this is the basis of the entire paper, this validation requires more rigorous approaches. These analyses should include quantification of NRP1-positive and negative microglia and non-microglial cells following TAM injection at time points consistent with all in vivo and in vitro analyses. The authors should also include quantification in the lysolecithin model.
- The conditional knock-out experiments lack important oil-treated groups to control for any off-target tamoxifen effects.
- In development, the authors show that there is initially a reduction in OPC proliferation in multiple brain regions, but the effect is no longer significant by P14. Is there any effect on OPC numbers, oligodendrocyte numbers, or myelination at older ages? It's assumed not, but these data would be helpful for interpretation of the developmental effects.
- Related to the point above, are effects on OPC proliferation balanced by changes in cell death such that overall numbers of OPCs are now unchanged? An assessment of OPC cell death would be informative.
- The underlying mechanism in this paper is that microglial NRP1 is necessary for regulating OPCs in the white matter. Thus, the data in Figure 2A-F showing NRP1 enrichment in F4/80 positive cells should be quantified.
- The authors state that they detected strong cellular NRP1 staining in only a subregion of the corpus callosum (CC, lines 223-224). However, representative images in (e.g. Fig 1A-F and Fig. 3O) appear to show very different areas of the corpus callosum. Also, the authors state that they detected NRP1 on a subset of microglia in the cerebellar white matter. This should be clarified and a better explanation of where representative images are acquired should be provided. If NRP1+ microglia differs depending on the region of the callosum and cerebellum, one would anticipate that OPC proliferation would be comparatively different in these subregions.
- In the description of Figure 3 M-P, authors state that they assessed changes in NRP1 expression between early postnatal development to adulthood, but they only show data up to P14.
- In Figure 4, the authors analyze the proliferative response of OPCs inside and outside of demyelinated lesions. The authors state that 'EdU incorporation into EYFP-negative, NG2+ OPCs around the lesions ("outside") of mg-Nrp1-cko was 3-fold lower than that in mg-Nrp1-cont lesions at both 3 and 7 dpl'. However, the provided data showed no effect outside the lesions at 7dpl. Also, it is unclear where the representative images are taken in Fig 4 F-I (i.e. inside or outside the lesion).
- In the author's model, NRP1+ microglia are in direct contact with OPCs to enhance PDGF AA-induced proliferation. Indeed, in line 451 authors state 'although OPCs were found in greater spatial proximity to NRP1+ microglia than to NRP1+ blood vessels', but no quantification is provided. To strengthen this argument, the authors should quantify data in Fig. 2 G-I and also include a microglial marker to ensure that NRP1+ cells in contact are, indeed, microglia. These analyses should be done in 3D and should also be performed in the demyelination model.
- Similar to the point above, the authors should quantify the percentage of total proliferating OPCs that are in contact with NRP1+ microglia versus not contacted or contacted by a NRP1- microglia.

It would be most convincing, if a greater proportion of the proliferating OPCs are in direct contact with NRP1+ microglia vs. NRP1-microglia.

- Could the authors also co-culture OPCs with NRP1+ and NRP1-negative microglia and assess the proliferative response of OPCs in this paradigm? Together with the above-mentioned experiments this would greatly strengthen the contact-dependent NRP1 role in microglia-induced OPC proliferation.
- The authors treated isolated primary mouse cortex OPC with NRP1-Fc in culture and found that this stimulation was sufficient to enhance PDGF AA-induced OPC proliferation (Figure 5). Given most effects were in white matter OPCs, it would be informative to do the same experiment in white matter-derived OPCs as well.
- A better description of the methods is required, including information on how images were acquired (z stacks, spacing of z-stacks, etc.), experimental blinding, and sex of animals.

Minor points:

- In the introduction, authors cite previous work showing PDGF AA-dependent proliferation of white matter OPCs, and refer to studies on the function of NRP1 in axon path finding. It remains unclear why NRP1 was selected as promising co-receptor that regulates the PDGF AA-dependent proliferative response of OPCs. A more detailed description should be included.
- Considering that most studies show DAPI IHC stain in blue and authors choose to do this as well in Fig. 4L-M, authors should consider a different pseudo color for the depiction of EdU signals throughout the manuscript, to avoid any confusion (e.g. Fig. 1, 3-6).
- In Fig. 1F and 5G, authors choose to show proliferation rates of gray matter OPCs next to the proliferative response of white matter OPCs. However, statistical comparisons are not being made between gray and white matter OPCs. It would be more appropriate to show the results on separate graphs.
- In Figure 6, the provided representative Western Blot images do not appear to represent a 1.5-fold increase in phospho-PDGFR α signal upon Nrp1-Fc stimulation.
- The densities of OPCs in Fig 6A,B appear very different.
- Figure titles should reflect major findings of the displayed data.
- Images showing NRP1 ablation in microglia would benefit from showing separated channels.
- The authors state that there is a dose-dependent effect of Nrp-Fc in Figure 5. However, while OPC proliferation in the 0 ug/ml Nrp1-Fc group is significantly different from the 0.4 and 2 ug/ml groups, 0.4 ug/ml group is not different than the higher concentrations. The conclusion should be refined.
- The arrangement of figures that include in vitro and in vivo data is confusing. Authors should consider rearranging and include better labeling.
- It is unclear how many technical and biological replicates have been used for the quantification of phospho-PDGFR α in Fig 6.
- In line 372 authors refer to "phagocytic microglia", however phagocytosis has not been assessed here. Authors only stained for lysosomal CD68, which, although increased in more phagocytic cells, is also present in non-phagocytic microglia. The authors should consider revising this terminology.

Reviewer #2:

Remarks to the Author:

Understanding the cellular and molecular mechanism that regulate the proliferation of oligodendrocyte precursors has important implications in both normal development and repair. In previous studies the authors provided evidence that the regulation of OPC proliferation depended on their environment. In the current study they have extended this analysis and propose that the enhancement of PDGF driven OPC proliferation is a consequence of the microglial expressed Neuropilin (Nrp1). In support of this hypothesis the authors show that deletion of Nrp1 from microglia compromises the proliferation of OPCs selectively in white matter and reduces OPC expansion after an LPC lesion. Furthermore, addition of exogenous Nrp1 enhances the response of

OPCs to PDGF driven proliferation in vitro. This is an interesting paper that raises a number of important issues. While the interpretation of the authors is quite well supported by the data there are a number of issues that remain unresolved.

1. In Fig 1 There appear to be a lot less OPCs in the presence of the Nrp1 antibody. This might reflect reduced proliferation or cell death and should be explored.

2. The majority of the assays are performed following the preparation of slice cultures. This is a potential confounding factor since the creation of the slice will itself induce microglial and astrocyte responses. The developmental studies would be more impactful if the analysis had been conducted in vivo.

3. The significance of the differential proliferative response mediated by Nrp1 is not clear. It would be important to determine if conditional deletion of Nrp1 in microglia had a lasting impact on myelination in white matter. Likewise, in the LPC lesions it would be helpful to know if there was an effect on remyelination. Lesion size is not a very specific assessment.

4. The authors imply that direct contact between microglia and OPCs mediate the Nrp1 effect. How this is mediated is not clear since it presumably requires "trans" interactions of the ligands. Alternatively, Nrp1 could be shed from microglia. It would be helpful to know if Nrp1 actually bound to OPCs or was internalized.

5. It is unclear why the residual expression of Nrp1 on axons and other cell types does not facilitate OPC proliferation. Is this a consequence of concentration or presentation?

We thank the reviewers for their comments. We have addressed each comment as detailed below and have indicated the changes made in the manuscript with blue font. In the following paragraphs, the reviewers' comments are copied in black font followed by our responses in blue font.

Reviewer #1

This study by Sherafat et al. provides evidence that microglial neuropilin-1 (NRP1) signaling enhances PDGF AA-mediated proliferation of oligodendrocyte precursor cells (OPCs) during development and after lysolecithin-induced demyelination. While the effects in development appear to be transient and relatively mild, during demyelination were more robust with effects on generation of new oligodendrocytes and remyelination. Mechanisms by which microglia regulate other glial cells is an understudied, but potentially very important, area of research. These findings define novel signaling between microglia and OPCs to modulate OPC proliferation, response to injury, and remyelination. Enthusiasm is generally high, but there are some concerns that require attention outlined below.

Major points:

1. To support the conclusions, the manuscript relies on the selective ablation of NRP1 in microglia. However, the data validating the microglia-specific NRP1 ablation are limited and are not quantified. Given this is the basis of the entire paper, this validation requires more rigorous approaches. These analyses should include quantification of NRP1-positive and negative microglia and non-microglial cells following TAM injection at time points consistent with all in vivo and in vitro analyses. The authors should also include quantification in the lysolecithin model.

We have quantified the proportion of YFP+ microglia that are Nrp1+ and Nrp1- at P5 and at 3 dpl and after LPC-induced demyelination after cre induction. The Nrp1 deletion efficiency was >99% of YFP+ cells in P5 and in adult corpus callosum after demyelination. We have also quantified the number of total microglia, astrocytes, and blood vessel profiles, and none of these cells or structures were altered in the cko (Supplemental Figure S4). We also show in a new graph that the density of microglia is comparable at 3 dpl after LPC injection in the corpus callosum of cont and cko mice (Figure 4F).

2. The conditional knock-out experiments lack important oil-treated groups to control for any off-target tamoxifen effects.

We have treated control and experimental groups with tamoxifen because we think this would detect any off-target tamoxifen effects.

3. In development, the authors show that there is initially a reduction in OPC proliferation in multiple brain regions, but the effect is no longer significant by P14. Is there any effect on OPC numbers, oligodendrocyte numbers, or myelination at older ages? It's assumed not, but these data would be helpful for interpretation of the developmental effects.

We show that developmental loss of Nrp1 from microglia caused a reduction of the OPC density that was detected through P14 but had no effect on MBP levels at P14 (Figure 3H and

text). We observed more long-lasting effects of *Nrp1* deletion on myelin repair. We have added new data on MBP immunofluorescence intensity and ultrastructural analysis at 28 dpl. These are presented in the new Figure 5. Please also see our response to Reviewer #2, comment 3.

4. Related to the point above, are effects on OPC proliferation balanced by changes in cell death such that overall numbers of OPCs are now unchanged? An assessment of OPC cell death would be informative.

We performed TUNEL assays on P5 *mg-Nrp1-cko* and *cont* sections and did not observe any increase in TUNEL+ cells in the corpus callosum of control or *cko* mice (Supplemental Figure S4).

5. The underlying mechanism in this paper is that microglial NRP1 is necessary for regulating OPCs in the white matter. Thus, the data in Figure 2A-F showing NRP1 enrichment in F4/80 positive cells should be quantified.

We have moved the original quantification of the percentage of microglial cells that were *Nrp1*-immunoreactive in the corpus callosum to Figure 2M with representative images in Figure 2J-L and additional images in Supplemental Figure S2.

6. The authors state that they detected strong cellular NRP1 staining in only a subregion of the corpus callosum (CC, lines 223-224). However, representative images in (e.g. Fig 1A-F and Fig. 3O) appear to show very different areas of the corpus callosum. Also, the authors state that they detected NRP1 on a subset of microglia in the cerebellar white matter. This should be clarified and a better explanation of where representative images are acquired should be provided. If NRP1+ microglia differs depending on the region of the callosum and cerebellum, one would anticipate that OPC proliferation would be comparatively different in these subregions.

We now provide images of *Nrp1* from comparable areas of the cortex and corpus callosum in the cingulum in the new Figure 2, J-L, and Supplemental Figure S2.

7. In the description of Figure 3 M-P, authors state that they assessed changes in NRP1 expression between early postnatal development to adulthood, but they only show data up to P14.

We have added a P30 time point in this dataset, which is now in Figure 2. We provide the data for P60 in Figure 4 when we describe the control for LPC-induced demyelination.

8. In Figure 4, the authors analyze the proliferative response of OPCs inside and outside of demyelinated lesions. The authors state that 'EdU incorporation into EYFP-negative, NG2+ OPCs around the lesions ("outside") of *mg-Nrp1-cko* was 3-fold lower than that in *mg-Nrp1-cont* lesions at both 3 and 7 dpl'. However, the provided data showed no effect outside the lesions at 7dpl. Also, it is unclear where the representative images are taken in Fig 4 F-I (i.e. inside or outside the lesion).

Because it was difficult to draw a clear line between outside and inside the lesion border, we have combined the counts of "outside" (area immediately surrounding the lesion) and "inside" the lesion and generated a new graph in Figure 4K. This shows that at both 3 and 7 dpl, there is a significant reduction in the number of proliferating OPCs in *mg-Nrp1-cko* mice.

9. In the author's model, NRP1+ microglia are in direct contact with OPCs to enhance PDGF AA-induced proliferation. Indeed, in line 451 authors state 'although OPCs were found in greater spatial proximity to NRP1+ microglia than to NRP1+ blood vessels', but not quantification is provided. To strengthen this argument, the authors should quantify data in Fig. 2 G-I and also include a microglial marker to ensure that NRP1+ cells in contact are, indeed, microglia. These analyses should be done in 3D and should also be performed in the demyelination model.

10. Similar to the point above, the authors should quantify the percentage of total proliferating OPCs that are in contact with NRP1+ microglia versus not contacted or contacted by a NRP1-microglia. It would be most convincing, if a greater proportion of the proliferating OPCs are in direct contact with NRP1+ microglia vs. NRP1-microglia.

We have repeated the quantification of EdU+ OPCs in P5 mg-Nrp1-cont and cko with a 3D analysis of the contact between EdU+ OPCs and microglia. We have also quantified the percentage of total OPCs that were contacting microglia and found that the proportion of OPCs that were contacting microglia was greater among the proliferating OPCs than among total OPCs, further suggesting that microglial contact facilitates OPC proliferation. The new data are included in Figure 3C-F. We could not perform a similar analysis on demyelinated lesions because the higher density of microglia and macrophages.

11. Could the authors also co-culture OPCs with NRP1+ and NRP1-negative microglia and assess the proliferative response of OPCs in this paradigm? Together with the above-mentioned experiments this would greatly strengthen the contact-dependent NRP1 role in microglia-induced OPC proliferation.

We have cocultured OPCs and microglia from cko and control mice and found that OPC proliferation in the presence of Nrp1-deficient microglia was significantly lower than that in the presence of microglia from control mice. Almost all of the microglia from cont mice expressed Nrp1 while none from cko mice had detectable Nrp1. While some OPCs that were apposed to control EYFP+ microglia had incorporated EdU, those that were apposed to cko EYFP+ microglia rarely had EdU, which indirectly suggesting that close proximity with microglial Nrp1 promotes OPC proliferation. However, we could not be certain that they had not been in contact prior to fixation, as these cells are highly motile in culture. Furthermore, it is recognized that isolated microglia in culture behave vastly differently from microglia in vivo. Therefore, we did not include this dataset in the revised manuscript, as we were uncertain of its relevance and significance.

12. The authors treated isolated primary mouse cortex OPC with NRP1-Fc in culture and found that this stimulation was sufficient to enhance PDGF AA-induced OPC proliferation (Figure 5). Given most effects were in white matter OPCs, it would be informative to do the same experiment in white matter-derived OPCs as well.

We did not perform the Nrp1-Fc experiments on white matter-derived OPCs because we did not detect any changes in OPC proliferation in white matter when we added Nrp1-Fc to slice cultures. Therefore, these experiments were done primarily to examine the mechanism underlying the ability of Nrp1-Fc to augment PDGF AA-mediated proliferation of gray matter OPCs.

13. A better description of the methods is required, including information on how images were acquired (z stacks, spacing of z-stacks, etc.), experimental blinding, and sex of animals.

We have included the information in the revised methods.

Minor points:

14. In the introduction, authors cite previous work showing PDGF AA-dependent proliferation of white matter OPCs, and refer to studies on the function of NRP1 in axon path finding. It remains unclear why NRP1 was selected as promising co-receptor that regulates the PDGF AA-dependent proliferative response of OPCs. A more detailed description should be included.

We have moved the rationale that was originally placed at the beginning of the Results section to the last paragraph of the Introduction and have modified the wording to make it clearer.

15. Considering that most studies show DAPI IHC stain in blue and authors choose to do this as well in Fig. 4L-M, authors should consider a different pseudo color for the depiction of EdU signals throughout the manuscript, to avoid any confusion (e.g. Fig. 1, 3-6).

We have changed the colors of the EdU label to a non-blue color as much as possible.

16. In Fig. 1F and 5G, authors choose to show proliferation rates of gray matter OPCs next to the proliferative response of white matter OPCs. However, statistical comparisons are not being made between gray and white matter OPCs. It would be more appropriate to show the results on separate graphs.

We have kept the data on the same graph and have performed comparisons between gray and white matter as well. The statistical details are included in the legends for Figure 1 and the new Figure 6.

17. In Figure 6, the provided representative Western Blot images do not appear to represent a 1.5-fold increase in phospho-PDGFR α signal upon Nrp1-Fc stimulation.

We have performed additional experiments and found that 30 minutes of incubation with Nrp1-Fc and PDGF AA led to a more robust increase in PDGFR α phosphorylation. We have replaced the original data with the new data in Figure 7.

18. The densities of OPCs in Fig 6A,B appear very different.

We have replaced them with images from areas with similar cell densities.

19. Figure titles should reflect major findings of the displayed data.

We have reworded the titles of the Figure panels to more closely match the data.

20. Images showing NRP1 ablation in microglia would benefit from showing separated channels.

We have included images from each confocal channel in Supplemental Figure S4, B-C.

21. The authors state that there is a dose-dependent effect of Nrp-Fc in Figure 5. However, while OPC proliferation in the 0 ug/ml Nrp1-Fc group is significantly different from the 0.4 and 2

ug/ml groups, 0.4 ug/ml group is not different than the higher concentrations. The conclusion should be refined.

We changed the wording in the Results section describing this to reflect more accurately what is presented in the graph.

22. The arrangement of figures that include in vitro and in vivo data is confusing. Authors should consider rearranging and include better labeling.

We have moved the slice culture data to Supplemental Figure S5 and taken it out of Figure 3. Figure 3 is now all in vivo developmental data.

23. It is unclear how many technical and biological replicates have been used for the quantification of phospho-PDGFRa in Fig 6.

This is in the legend for the figure (new Figure 7).

24. In line 372 authors refer to “phagocytic microglia”, however phagocytosis has not been assessed here. Authors only stained for lysosomal CD68, which, although increased in more phagocytic cells, is also present in non-phagocytic microglia. The authors should consider revising this terminology.

We have changed the word "phagocytic" to "activated".

Reviewer #2 (Remarks to the Author):

Understanding the cellular and molecular mechanism that regulate the proliferation of oligodendrocyte precursors has important implications in both normal development and repair. In previous studies the authors provided evidence that the regulation of OPC proliferation depended on their environment. In the current study they have extended this analysis and propose that the enhancement of PDGF driven OPC proliferation is a consequence of the microglial expressed Neuropilin (Nrp1). In support of this hypothesis the authors show that deletion of Nrp1 from microglia compromises the proliferation of OPCs selectively in white matter and reduces OPC expansion after an LPC lesion. Furthermore, addition of exogenous Nrp1 enhances the response of OPCs to PDGF driven proliferation in vitro. This is an interesting paper that raises a number of important issues. While the interpretation of the authors is quite well supported by the data there are a number of issues that remain unresolved.

1. In Fig 1 There appear to be a lot less OPCs in the presence of the Nrp1 antibody. This might reflect reduced proliferation or cell death and should be explored.

To assess whether anti-Nrp1 antibody had caused cell death, we performed TUNEL labeling of slice cultures treated with control goat IgG and goat anti-Nrp1 antibody. We saw a similar number of TUNEL+ cells in cultures treated with either condition. Thus, we attribute the reduced density of OPCs in the slice culture to reduced proliferation. The new data are included in Fig 1, G-H.

2. The majority of the assays are performed following the preparation of slice cultures. This is a potential confounding factor since the creation of the slice will itself induce microglial and astrocyte responses. The developmental studies would be more impactful if the analysis had been conducted in vivo.

We have performed in vivo developmental studies in which we demonstrate reduced OPC proliferation in P5 corpus callosum and cerebellar white matter (original Figure 3). We have tried to improve the presentation of the in vivo and vitro experiments. To avoid confusion, we have moved the slice culture experiments using knockout slices to Supplemental Figure 5, and the new Figure 3 now consists entirely of in vivo data. We wished to include the slice culture experiments to demonstrate that the specific proliferative response of OPCs to PDGF-AA is affected by microglial Nrp1 deletion, since the in vivo experiments could only measure general OPC proliferation. We were careful to obtain counts only from middle z-levels of the slices to avoid activated microglia and astrocytes near the surface.

3. The significance of the differential proliferative response mediated by Nrp1 is not clear. It would be important to determine if conditional deletion of Nrp1 in microglia had a lasting impact on myelination in white matter. Likewise, in the LPC lesions it would be helpful to know if there was an effect on remyelination. Lesion size is not a very specific assessment.

We have extensively revised the manuscript to include the effects of microglial Nrp1 cko on remyelination. We have measured MBP fluorescence intensity at 28 dpl after demyelination and show that the level of MBP is 39% lower in mg-Nrp1-cko corpus callosum compared with that in control mice. Furthermore, we have performed electron microscopy on the remyelinating corpus callosum at 28 dpl and show evidence of delayed repair process in the cko, such as more unmyelinated axons, more prominent glial cells surrounded by axons with varying myelin morphologies, and an abundance and persistence of swollen glial processes. These observations suggest that deletion of Nrp1 from microglia have a long-lasting effect on myelin repair as a consequence of its effect on OPC proliferation. The new findings are included in the new Figure 5. We have also included data on the developmental effects showing that microglial deletion of Nrp1 did not have any effect on MBP immunofluorescence intensity at P14 (Figure 3H), although OPC density remained lower than control mice.

4. The authors imply that direct contact between microglia and OPCs mediate the Nrp1 effect. How this is mediated is not clear since it presumably requires "trans" interactions of the ligands. Alternatively, Nrp1 could be shed from microglia. It would be helpful to know if Nrp1 actually bound to OPCs or was internalized.

We incubated dissociated cultures of OPCs with control-Fc or Nrp1-Fc and show that Nrp1-Fc co-clustered with PDGFR α on OPCs at 36°C but not at 4°C, and that control-Fc did not cluster into discrete puncta. We could not assess from these results whether the colocalized puncta had been internalized. Nevertheless, these observations support the notion that Nrp1-Fc binds to PDGFR α on OPCs and exert a functional effect on the receptor. The results are shown in Supplemental Figure 6.

5. It is unclear why the residual expression of Nrp1 on axons and other cell types does not facilitate OPC proliferation. Is this a consequence of concentration or presentation?

Nrp1 was robustly expressed on axons at E18, but the level was significantly lower by P5 when OPC proliferation is near its maximum. While it is possible that axonal Nrp1 affects earlier OPC proliferation, we focused on the role of microglial Nrp1 in postnatal OPC proliferation in this study, which seemed to be more positively corrected with the expression of Nrp1 on amoeboid microglia. Furthermore, we did not see an upregulation of Nrp1 on axons or vascular cells after LPC-induced demyelination when we saw a prominent effect on microglial deletion of Nrp1 on OPC proliferation. Since proliferating OPCs were not as closely associated with Nrp1+ blood vessels, compared to the close proximity to Nrp1+ microglia, we analyzed the role of microglia and did not examine the role of Nrp1 on blood vessels in this study. It is possible that deletion of Nrp1 from both microglia and vascular cells would have a more prominent inhibitory effect on OPC proliferation. This is already in the discussion in the second paragraph on page 19.

Reviewers' Comments:

Reviewer #1:

Remarks to the Author:

The authors have done a terrific job addressing most concerns. There are just two points that require more attention:

1. The inclusion of new analyses of OPC cell death is useful. The authors state on line 333-334 that "The density of total OPCs in the corpus callosum of mg-Nrp1-cko mice was 1.7-fold lower than that in control mice at P5 and remained 1.4-fold lower at P14." This quantification should be included in the figures.
2. The inclusion of new 3D reconstructions of OPC-microglia contact in vivo in revised Figure 3 is appreciated. Could the authors include the raw fluorescence images and more zoomed-in images to better appreciate the intercellular interactions? It is also recommended to more clearly define what is meant by contact. Do the cells have to be within a certain distance of an OPC before it is defined as "contact" vs. close proximity? Can the authors be sure this is contact at the level of confocal light microscopy in vivo?

Reviewer #2:

Remarks to the Author:

In response to the previous round of reviews the authors have made major efforts to address the issues raised. The paper is considerably stronger and more complete as a consequence. The additional studies with cell death analysis and refinement of the quantitation of the proliferation results has provide increased rigor. The developmental studies are also very much stronger. This paper is an interesting and novel contribution to the growing literature around the role of microglia in regulating the generation of myelinating cells in the vertebrate CNS. The data is of a high quality and strongly supports the interpretation of authors. There remain a few issues that should be addressed prior to publication.

There are a few areas that might be considered.

1. It was somewhat surprising that the authors chose not to include the culture studies (comment 11 reviewer 1). While it is correct that contact may be difficult to discern with highly motile cells, and microglia in vitro may be different, the fact that they have differences in cell proliferation between control and mutant cultures is at least worthy of a mention. It is some of the strongest data for specific cell associations.
- 2 There are some concerns regarding the EM data showing delayed remyelination in the mg-Nrp-1 cko animals. The data as presented is not very convincing. It might be better to identify an axon cluster or a few axons that typify the difference in the rate of repair between the phenotypes.
3. The proliferation of OPCs after LPC lesion appears to be higher at 3 days post-lesion than at 7 days post lesion. This is a surprise since several published studies suggest that OPC proliferation is higher at pd7 than pd3. This raises the question of what proportion of the NG2+ cells are OPCs and this is worthy of comment.
4. Figure 8 seems unnecessary. This is not a very complicated story and probably does not justify an overview figure.

We thank the reviewers for the additional comments and suggestions. We have added the suggested changes in the revised manuscript and have marked the changes in blue font. Below is a detailed response to the reviewers' comments.

Reviewer #1:

The authors have done a terrific job addressing most concerns. There are just two points that require more attention:

1. The inclusion of new analyses of OPC cell death is useful. The authors state on line 333-334 that "The density of total OPCs in the corpus callosum of mg-Nrp1-cko mice was 1.7-fold lower than that in control mice at P5 and remained 1.4-fold lower at P14." This quantification should be included in the figures.

We have included the results of an updated quantification of OPC density in similar regions of the corpus callosum and show the new data in Figure 3H. The new quantification revealed that while there was a 1.7-fold reduction in the density of OPCs in mg-Nrp1-cko mice at P5, the difference was no longer detected at P14. The text has been updated to reflect this.

2. The inclusion of new 3D reconstructions of OPC-microglia contact in vivo in revised Figure 3 is appreciated. Could the authors include the raw fluorescence images and more zoomed-in images to better appreciate the intercellular interactions? It is also recommended to more clearly define what is meant by contact. Do the cells have to be within a certain distance of an OPC before it is defined as "contact" vs. close proximity? Can the authors be sure this is contact at the level of confocal light microscopy in vivo?

We have included a new Supplemental Figure 5 with a series of consecutive z-sections from confocal stacks to demonstrate what we define as contact or no contact with microglia.

Reviewer #2:

In response to the previous round of reviews the authors have made major efforts to address the issues raised. The paper is considerably stronger and more complete as a consequence. The additional studies with cell death analysis and refinement of the quantitation of the proliferation results has provide increased rigor. The developmental studies are also very much stronger.

This paper is an interesting and novel contribution to the growing literature around the role of microglia in regulating the generation of myelinating cells in the vertebrate CNS. The data is of a high quality and strongly supports the interpretation of authors. There remain a few issues that should be addressed prior to publication.

There are a few areas that might be considered.

1. It was somewhat surprising that the authors chose not to include the culture studies (comment 11 reviewer 1). While it is correct that contact may be difficult to discern with highly motile cells, and microglia in vitro may be different, the fact that they have differences in cell proliferation between control and mutant cultures is at least worthy of a mention. It is some of the strongest data for specific cell associations.

We have included the data on OPC proliferation in OPC-microglial cocultures using microglia from mg-Nrp1-cont and mg-Nrp1-cko mice in Supplemental Figure 6B.

2 There are some concerns regarding the EM data showing delayed remyelination in the mg-Nrp-1 cko animals. The data as presented is not very convincing. It might be better to identify an axon cluster or a few axons that typify the difference in the rate of repair between the phenotypes.

We have included a different region for mg-Nrp1-cont corpus callosum that may better represent more significant repair in mg-Nrp1-cont (fl/+) mice compared to the cko (fl/fl) mice in Figure 5, J and K. Additional EM images have been moved to Supplemental Figure 7.

3. The proliferation of OPCs after LPC lesion appears to be higher at 3 days post-lesion than at 7 days post lesion. This is a surprise since several published studies suggest that OPC proliferation is higher at pd7 than pd3. This raises the question of what proportion of the NG2+ cells are OPCs and this is worthy of comment.

In our earlier study, we noted that there is a significantly greater proliferative activity of OPCs immediately surrounding the lesion at 2 dpl than 7 dpl, whereas within the lesion core, a greater number of proliferating OPCs are detected at 7 dpl. This likely reflects migration of proliferating OPCs from around the lesion to the lesion core. Since we included in our analysis the area immediately surrounding the core demyelinated area, the higher proliferation of OPCs at 3 dpl was expected. The data also suggests that the early proliferative response of OPCs to acute demyelination is compromised by loss of Nrp1 from microglia. We have also cited a study which describes 3 dpl as the period of recruitment (proliferation and migration) in response to lysolecithin-induced demyelination in the corpus callosum.

4. Figure 8 seems unnecessary. This is not a very complicated story and probably does not justify an overview figure.

Since we are permitted to have 10 display items in the main text, we have decided to keep the summary schematic in Figure 8, as we think it illustrates the novel concept we are proposing from our data.